# FlowKV: Enhancing Multi-Turn Conversational Coherence in LLMs via Isolated Key-Value Cache Management

## Abstract

Large Language Models (LLMs) are increasingly deployed in multi-turn conversational applications, where the management of the Key-Value (KV) Cache presents a significant bottleneck. The linear growth of the KV Cache with dialogue history imposes substantial computational costs, and existing eviction strategies often degrade performance by repeatedly compressing early conversational context, leading to information loss and context forgetting. This paper introduces FlowKV, a novel **multi-turn isolation mechanism** for KV Cache management, which can be applied to any KV Cache compression method without training. FlowKV's core innovation is a multi-turn isolation mechanism that preserves the accumulated compressed KV cache from past turns. Compression is then strategically applied only to the newly generated KV pairs of the latest completed turn, effectively preventing the re-compression of older context and thereby mitigating catastrophic forgetting. Our results demonstrate that FlowKV consistently and significantly outperforms baseline strategies in maintaining instruction-following accuracy and user preference retention from 10.90% to 75.40%, particularly in later conversational turns.

## 1 Introduction

Large Language Models (LLMs) have achieved remarkable success in a wide range of natural language understanding and generation tasks, powering applications from open-domain dialogue systems to complex instruction following (Raffel et al., 2020; Brown et al., 2020; Chowdhery et al., 2022; Tay et al., 2022; Touvron et al., 2023a;b; Dubey et al., 2024; Guo et al., 2025). However, as these models are increasingly deployed in multi-turn conversational scenarios, new challenges emerge regarding their efficiency and ability to maintain coherent, contextually relevant responses over extended interactions.

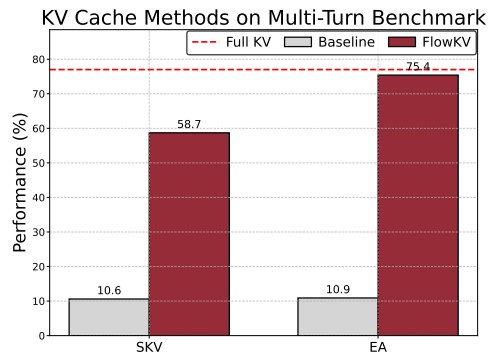

Figure 1: KV cache compression on PrefEval. (#SKV: SnapKV, #EA: ExpectedAttention).

The complexity of managing information flow in such multi-turn settings is evident in the attention patterns of LLMs. Figure 2 visualizes an attention heatmap from a representative 3-turn dialogue. The heatmap reveals several crucial characteristics: sustained attention to initial system prompts (SYS) throughout the conversation, strong local attention within each conversational turn, and, critically, significant cross-turn dependencies. For instance, **responses not only attend to their immediate queries but also to previous queries and responses** (e.g., the Turn 2 Response attending to T1Q and T1R). Furthermore, subsequent user queries (e.g., T2Q and T3Q) demonstrate an evolving understanding by referencing earlier queries and the system prompt. These intricate attention patterns, highlighting the model's reliance on both recent and distant context, directly underscore the challenges in efficiently managing the conversational history.

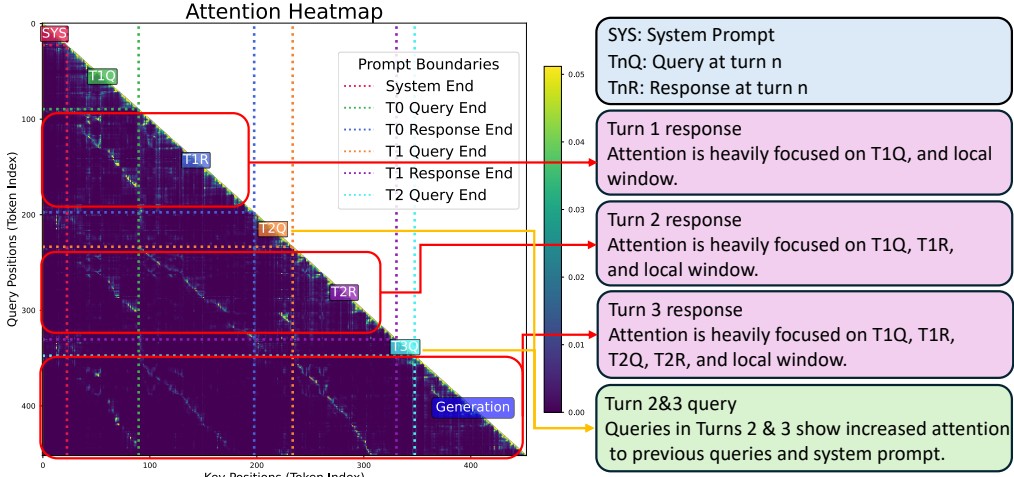

Figure 2: Attention heatmap illustrating token-level focus during a 3-turn conversational interaction. The Y-axis represents query token positions, while the X-axis shows key token positions. Key observations (highlighted on the right) include: (1) Responses (e.g., Turn 1 Response) heavily focus on their **corresponding query (e.g., T1Q) and the local context (local window)**. (2) As the dialogue progresses, responses (e.g., Turn 2 and 3 Responses) attend to an increasing span of **historical context**, including previous queries and responses (e.g., T1Q, T1R, T2Q, T2R). (3) Later queries (Turn 2 & 3 Queries) exhibit increased attention to both previous queries and the initial system prompt, indicating an evolving contextual understanding. These patterns underscore the complex, long-range dependencies managed by the attention mechanism in multi-turn dialogues.

This challenge of efficiently managing extensive conversational history, so critical for maintaining coherence as highlighted by the attention patterns, is primarily rooted in the operational characteristics of the KV Cache within the attention mechanism. While the KV Cache enables efficient autoregressive generation by storing intermediate representations, its memory footprint grows linearly with the length of the conversation history (Xiao et al., 2023). This not only imposes significant computational and storage costs but also limits the practical deployment of LLMs in real-world, resource-constrained environments.

To address these challenges, a variety of KV Cache eviction strategies have been proposed (Xiao et al., 2023; Zhang et al., 2023; Liu et al., 2025b; Li et al., 2024a; Liu et al., 2025a; Cai et al., 2024; Yang et al., 2024b). Despite their promise, the effectiveness of these methods in multi-turn dialogue settings remains underexplored. In particular, it is unclear how different compression strategies impact the model's ability to follow instructions, retain user preferences, and maintain long-term dependencies across multiple conversational turns.

In this work, we conduct a comprehensive empirical study of KV Cache eviction strategies in multi-turn dialogue scenarios. We benchmark several state-of-the-art techniques—including SnapKV (Li et al., 2024a), StreamingLLM (Xiao et al., 2023), ExpectedAttention (Jegou et al., 2024), and ChunkKV (Liu et al., 2025b)—on two representative datasets: Multi-IF (He et al., 2024), which evaluates instruction following across turns, and PrefEval (Zhao et al., 2025), which assesses user preference retention. Figure 1 shows these methods' performance on PrefEval, highlighting the KV cache compression method still have a large gap with the full KV cache. Furthermore, we introduce **FlowKV**, a multi-turn isolation mechanism designed to mitigate information loss and context forgetting under aggressive compression. This training-free approach is compatible with any existing KV Cache compression method and demonstrates significant improvements, for instance, boosting the average Instruction Following Rate (IFR) by over 20% on Multi-IF and increasing user preference adherence on PrefEval from as low as 10.90% to 75.40% with LLaMA-3.1-8B.

Our results demonstrate that FlowKV consistently outperforms baseline strategies, especially in later turns where context accumulation and compression effects are most pronounced. Through extensive quantitative and qualitative analyses, we provide new insights into the trade-offs between efficiency and conversational ability in LLMs, paving the way for more robust and scalable dialogue systems.

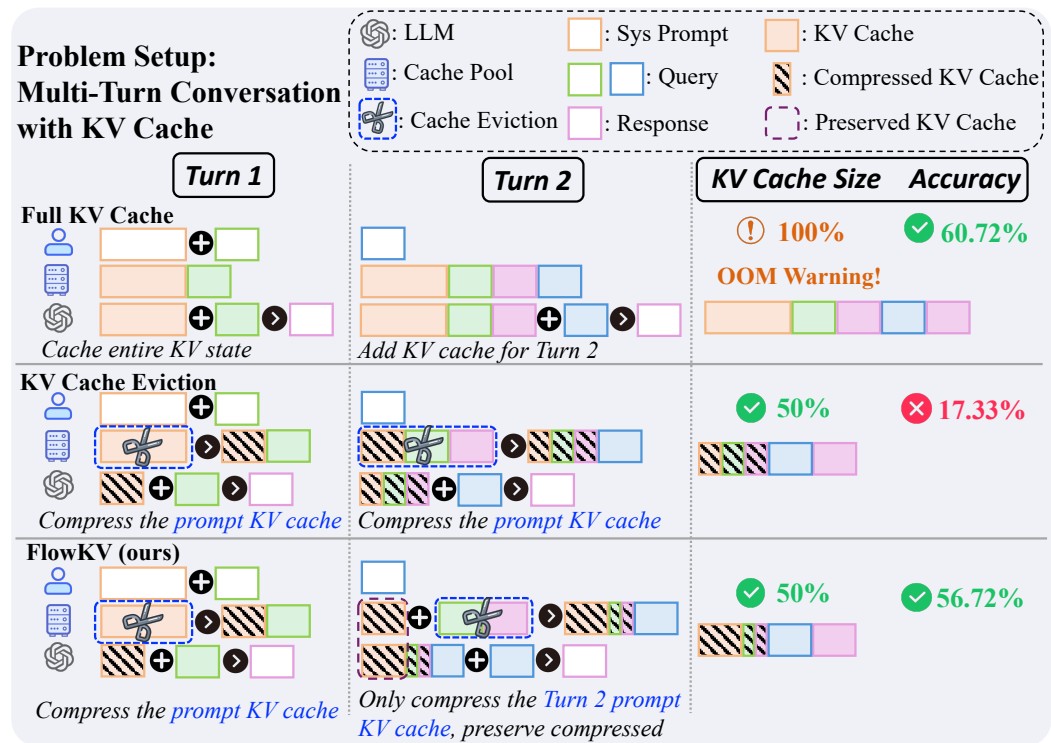

Figure 3: Illustration of KV cache dynamics across three management strategies in a two-turn conversational setting. Each turn consists of a system prompt (Sys Prompt), user query (Query), and model response (Response), which contribute to the KV cache. **Top Row (Full KV Cache):** All KV states are retained, achieving high accuracy (60.72%) but leading to OOM errors. **Middle Row (KV Cache Eviction):** Prompt-related KV cache is compressed each turn. This reduces cache size by 50% but results in a severe accuracy drop to 17.33%. **Bottom Row (FlowKV - Ours):** Our proposed **FlowKV** method also reduces cache size by 50% by selectively compressing only the current turn's prompt-related KV cache while preserving already compressed states from previous turns. This strategy effectively mitigates OOM issues while maintaining a high accuracy of 56.72%, significantly outperforming simple eviction.

## 2 FLOWKV: MULTI-TURN CONTEXT ISOLATION

### 2.1 PRELIMINARIES: KV CACHE DYNAMICS IN MULTI-TURN CONVERSATIONS

The KV cache is a fundamental component in Transformer-based LLMs (Vaswani et al., 2017), designed to enhance the efficiency of autoregressive token generation. It stores the computed key and value vectors for all preceding tokens in a sequence, thereby obviating the need for their recomputation at each subsequent generation step.

In a multi-turn conversational setting, the context grows with each turn. Let $P_{sys}$ denote the initial system prompt. For turn $t$, let $Q_t$ be the user query and $R_t$ be the model's response, $C_t$ be the KV cache pool at turn $t$. Under a standard **Full KV Cache** strategy, the entire conversational history is maintained. The KV cache accumulation can be described as follows:

The **Initial State** involves the system prompt.

$$C_0 = \text{KV}(P_{sys}) \tag{1}$$

For **Turn 1**, the model processes the first user query $Q_1$ using the context of $P_{sys}$. The KV cache for the prompt segment leading to the generation of $R_1$ is:

$$C_1 = \text{KV}(P_{sys} \oplus Q_1) \tag{2}$$

where $\oplus$ denotes sequence concatenation. After $R_1$ is generated, its KV pairs are added. The total KV cache state after Turn 1, $C_1$, becomes:

$$C_1' = C_1 \oplus \text{KV}(R_1) = \text{KV}(P_{sys} \oplus Q_1 \oplus R_1) \tag{3}$$

This $C_1'$ is carried forward.

For any subsequent **Turn** $t$ ($t > 1$), let $H_{t-1}$ be the complete conversational history up to the end of turn $t-1$:

$$H_{t-1} = P_{sys} \oplus Q_1 \oplus R_1 \oplus \cdots \oplus Q_{t-1} \oplus R_{t-1} \tag{4}$$

The KV cache state from the previous turn is $C_{t-1} = \text{KV}(H_{t-1})$. To generate $R_t$, the model processes $Q_t$ using the context $H_{t-1}$. The prompt KV cache for generating $R_t$ is:

$$C_t = \text{KV}(H_{t-1} \oplus Q_t) \tag{5}$$

After $R_t$ is generated, the total KV cache state $C_t$ becomes:

$$C_t' = C_t \oplus \text{KV}(R_t) = \text{KV}(H_{t-1} \oplus Q_t \oplus R_t) \tag{6}$$

This cumulative growth means $C_t$ includes all KV pairs from $P_{sys}$, all $Q_i$ and $R_i$ for $i \leq t$. While this Full KV Cache approach ensures maximum contextual information for the model, its linear growth with conversation length leads to substantial memory demands and computational overhead, presenting a critical challenge for deploying LLMs in extended dialogues.

In contrast to the Full KV Cache, many traditional KV cache eviction strategies aim to reduce memory by compressing the conversational history accumulated up to the current turn, before processing the current user query. A common approach, similar to mechanisms found in frameworks like NVIDIA's kvpress (Jegou et al., 2024), while leaving the *KV cache of the current turn's query uncompressed* for better performance.

At **Turn 1**: The KV cache of the system prompt, $P_{sys}$, is compressed:

$$C_1 = \mathcal{F}(\text{KV}(P_{sys})) \oplus \text{KV}(Q_1) \tag{7}$$

The response $R_1$ is then generated using the context formed by this compressed system prompt and the uncompressed KV cache of the first query. where $\mathcal{F}(\cdot, b)$ denotes a generic compression operator that reduces the input to a budget $b$. After $R_1$ is generated, the total KV cache state $C_1$ becomes:

$$C_1' = \mathcal{F}(\text{KV}(P_{sys})) \oplus \text{KV}(Q_1 \oplus R_1) \tag{8}$$

At this point, the original $P_{sys}$ has been subjected to one compression operation.

At **Turn 2**: The $C_1'$ will be compressed before the second turn response:

$$C_2 = \mathcal{F}(C_1') \oplus \text{KV}(Q_2) \tag{9}$$

The response $R_2$ is generated using the context formed by this newly compressed history $C_1$ and the uncompressed KV cache of the second query.

As shown in Equation (9), the original system prompt information $P_{sys}$ (first compressed to $\mathcal{F}(P_{sys})$) is part of the input to a subsequent compression operation $\mathcal{F}(C_1')$. Thus, $P_{sys}$ has effectively been compressed twice.

This pattern continues. After $T$ turns, the KV cache corresponding to the initial system prompt, $P_{sys}$, would have been subjected to $T$ nested compression operations on the path it contributes to the final compressed history. And $KV(Q_i)$ and $KV(R_i)$ for $i \leq t$ will be compressed $T - i$ times.

This repeated, nested compression of early conversational history is a primary cause of severe information degradation and context forgetting in such eviction strategies, leading to a rapid decline in coherence and task performance as the dialogue lengthens.

## 2.2 FLOWKV OPERATIONAL MECHANISM

To address the challenge of information degradation from repeated compressions in naive KV cache eviction strategies (as discussed in Section 2.1), we introduce **FlowKV**. Designed as a *multi-turn isolation mechanism*, FlowKV's core principle is to preserve the integrity of the already accumulated cache pool from previous turns ($C_{t-1}$), which contains previously compressed system prompts and

uncompressed queries and responses. This approach mitigates catastrophic forgetting by avoiding re-compression of historical data. Furthermore, FlowKV is a general framework that can be integrated with any existing KV cache compression methods.

The operational mechanism of FlowKV is best understood by considering its behavior at each conversational turn $t$, and is visually contrasted with other methods in Figure 3 (Bottom Row).

**Turn 1 (Identical to Traditional Eviction):** For the first turn ($t = 1$), FlowKV mirrors the behavior of the traditional eviction strategy described earlier.

$$C_1 = \mathcal{F}(\text{KV}(P_{sys})) \oplus \text{KV}(Q_1) \tag{10}$$

The response $R_1$ is then generated using the context formed by this compressed system prompt and the uncompressed KV cache of the first query. The total cache pool state after Turn 1, $C_1$, which is carried forward, incorporates the compressed system prompt, the uncompressed query, and the uncompressed response:

$$C_1' = \mathcal{F}(\text{KV}(P_{sys})) \oplus \text{KV}(Q_1 \oplus R_1) \tag{11}$$

At **Turn 2**: When new inputs for the current turn query $Q_2$ is received, FlowKV focuses its compression effort on uncompressed part of $C_1$:

$$C_2 = \mathcal{F}(C_1') \oplus \text{KV}(Q_2) \tag{12}$$

Where $(C_1')$ is cache pool that apply FlowKV isolation mechanism, to be specific that the $\mathcal{F}(\text{KV}(P_{sys}))$ will preserved and only $KV(Q_1 \oplus R_1)$ will be compressed. Compare the equation (8) and (9), we can see that only the $P_{sys}$ is compressed twice, while $Q_1$ and $R_1$ are compressed once. With FlowKV, the $P_{sys}$, $KV(Q_i)$ and $KV(R_i)$ for $i \leq t$ will be have better preservation. This turn-by-turn isolation is the cornerstone of FlowKV. By not subjecting the accumulated history to repeated or joint compression, FlowKV minimizes the degradation of older contextual cues. For more details about FlowKV, please refer to Appendix D and E.

## 3 EXPERIMENTS DESIGN

### 3.1 EXPERIMENTAL SETUPS

This section will introduce the experimental setups, including the datasets, models, and evaluation environment.

**Datasets**  To evaluate the performance of LLMs under multi-turn conversation with different KV Cache compression strategies, we assess the Multi-IF (He et al., 2024) and the PrefEval (Zhao et al., 2025) datasets. Multi-IF focuses on evaluating the continuous instruction-following ability of LLMs across multiple turns of conversation, and directly identifies the influence of compression strategies on long-range context dependence and instruction execution accuracy. Besides, PrefEval provides diverse and challenging conversation prompts with hidden user preferences. It can be used to evaluate the impact of compression on the model's ability to maintain conversation coherence and follow user preferences in complex interactions.

**Evaluation Metrics**  In terms of the Multi-IF dataset, we use Instruction Following Rate (IFR) to evaluate the model's performance when facing multiple instructions:

$$\text{IFR} = \frac{\text{SPA} + \text{SIA} + \text{LPA} + \text{LIA}}{4} \tag{13}$$

where SPA refers to Strict Prompt-level Accuracy, SIA denotes Strict Instruction-level Accuracy, LPA means Loose Prompt-level Accuracy, and LIA signifies Loose Instruction-level Accuracy. Instruction-level calculates the percentage of followed instructions out of the total instructions within the same prompt, while Prompt-level assesses if the reply follows all instructions in the prompt, failing entirely if any single instruction is not followed. Strict adherence demands exact correspondence with the instruction, whereas Loose relaxes this by removing empty lines and insignificant symbols, among other things.

In terms of PrefEval, we use GPT-4o as a judge. For each model response, we provide GPT-4o with the response and the explicit user preference from the dataset and ask it if the response follows that preference. The User Preference Following Rate is then just the percentage of "yes" answers from GPT-4o.

**Models**   We conduct experiments on two LLMs, including LLaMA-3.1-8B-Instruct (Grattafiori et al., 2024) and Qwen-2.5-7B-Instruct (Yang et al., 2024a).

**KV Cache Compression Methods**   We selected the following KV cache compression methods for a comprehensive investigation of their effects: SnapKV (Li et al., 2024a), StreamingLLM (Xiao et al., 2023), ExpectedAttention (Jegou et al., 2024), ChunkKV (Liu et al., 2025b) and FlowKV. For below experiments, baseline means the method without FlowKV.

**Strict Budget Adherence**   We emphasize that FlowKV operates under a strict, fixed memory budget identical to the baselines. As detailed in Appendix E, we dynamically adjust the compression ratio of the new turn to ensure the total cache size (Frozen History + New Turn) never exceeds the pre-defined limit (e.g., 50% of Full KV).

**Evaluation Environments**   We apply the kvpress (Jegou et al., 2024) library to load models and KV Cache compression methods and evaluate their performance using a server with NVIDIA A40 GPUs. All experiments are conducted three times and the average results are reported.

## 3.2 PERFORMANCE COMPARISON ON MULTI-IF

In this section, we will present and analyze the instruction following the performance of FlowKV and the baseline on the Multi-IF dataset.

As shown in Table 1, during the initial turn of the conversation, the core isolation mechanism of FlowKV is not yet engaged due to the absence of prior conversation history. However, starting from the second turn and continuing into the third turn, FlowKV outperforms the baseline as the contextual information accumulates and the compression mechanism begins to exert a significant effect. FlowKV achieves state-of-the-art IFR performance in the subsequent conversation turns for the current configuration. On average, there is a performance improvement of over 20%. With FlowKV, SnapKV and ExpectedAttention exhibit minimal performance degradation compared to FullKV, demonstrating that FlowKV's multi-turn isolation strategy can maximally preserve information flow in multi-turn dialogues. For StreamingLLM, this method discards intermediate parts of the KV Cache, which inherently leads to the loss of intermediate information in multi-turn dialogues, resulting in significant performance degradation.

To gain a more comprehensive understanding of the robustness of FlowKV under different compression ratios, we further examined the performance of each method across a range of compression ratios from 0.1 to 0.9, where a higher compression ratio corresponds to a smaller KV cache size. As shown in Figure 4, we first compare the overall trends of the yellow line (SnapKV) and the green line (ExpectedAttention). It is evident that the performance of all

Table 1: Overall IFR Comparison on the Multi-IF Dataset. All results are reported with a KV compression ratio of 0.5. For FlowKV, delta is the absolute value. # FKV: FullKV, SKV: SnapKV, SLM: StreamingLLM, EA: ExpectedAttention, CKV: ChunkKV

| KV Method | Strategy | IFR↑ | | |
|---|---|---|---|---|
| | | Turn 1 | Turn 2 △ | Turn 3 △ |
| **LLaMA-3.1-8B-Instruct** | | | | |
| FKV | - | 73.41% | 64.49% | 56.62% |
| SKV | Baseline | 76.15% | 37.08% | 29.39% |
| | FlowKV | 76.15% | **61.93%** $_{+24.85}$ | **54.95%** $_{+25.56}$ |
| SLM | Baseline | 72.78% | 33.94% | 28.94% |
| | FlowKV | 72.78% | **39.06%** $_{+5.12}$ | **41.58%** $_{+12.64}$ |
| EA | Baseline | 76.05% | 36.28% | 30.48% |
| | FlowKV | 76.05% | **64.89%** $_{+28.61}$ | **55.36%** $_{+24.88}$ |
| CKV | Baseline | 70.47% | 12.56% | 16.49% |
| | FlowKV | 70.47% | **52.83%** $_{+40.27}$ | **50.15%** $_{+33.66}$ |
| **Qwen-2.5-7B-Instruct** | | | | |
| FKV | - | 76.30% | 60.72% | 51.19% |
| SKV | Baseline | 76.49% | 17.33% | 21.96% |
| | FlowKV | 76.49% | **56.72%** $_{+39.39}$ | **49.67%** $_{+27.71}$ |
| SLM | Baseline | 76.47% | 17.31% | 21.08% |
| | FlowKV | 76.47% | **36.82%** $_{+19.51}$ | **35.29%** $_{+14.21}$ |
| EA | Baseline | 75.52% | 17.62% | 22.00% |
| | FlowKV | 75.52% | **50.62%** $_{+33.00}$ | **39.25%** $_{+17.25}$ |
| CKV | Baseline | 73.19% | 18.47% | 21.51% |
| | FlowKV | 73.19% | **47.27%** $_{+28.80}$ | **43.55%** $_{+22.04}$ |

Table 2: Overall User Preference Following Rate Comparison on the PrefEval Dataset. All results are reported with a 0.5 KV compression ratio.

| Strategy | SKV | SLM | EA | CKV |
|---|---|---|---|---|
| LLaMA-3.1-8B-Instruct Full KV: 77.00% ↑ | | | | |
| Baseline | 10.60% | 9.80% | 10.90% | 6.70% |
| **FlowKV** | **58.70%** | **24.40%** | **75.40%** | **38.80%** |
| Qwen-2.5-7B-Instruct Full KV: 55.90% ↑ | | | | |
| Baseline | 11.80% | 11.60% | 10.60% | 10.30% |
| **FlowKV** | **33.80%** | **16.80%** | **29.80%** | **26.40%** |

methods decreases as the compression ratio increases, which is consistent with the expected information loss caused by higher compression. Next, by comparing FlowKV (triangle markers) with the baseline, we observe that FlowKV consistently outperforms the baseline in both Turn 2 and Turn 3. Notably, FlowKV achieves substantial performance improvements across all compression ratios. The gains are particularly pronounced at lower compression ratios (0.1–0.4), while at higher compression ratios (0.5–0.9), FlowKV still provides significant improvements, although the margin is relatively smaller compared to the low compression regime. These results indicate that FlowKV can robustly preserve the information flow across multiple dialogue turns, irrespective of the compression ratio. More visualizaiton for LLaMA-3.1-8B-Instruct and Qwen-2.5-7B-Instruct can be found in B.3.

## 3.3 PERFORMANCE COMPARISON ON PREFEVAL

The Table 2 shows the performance of different compression methods on PrefEval with a compression ratio of 0.5. Firstly, it can be seen that the baselines' scores are very low comparing to the Full KV Cache, indicating that traditional KV Cache compression causes severe performance loss in multi-turn dialogue human preference benchmarks. When FlowKV is added, the performance improves significantly. For the LLaMA-3.1-8B-Instruct model with the ExpectedAttention method, the performance increased from 10.90% to 75.40%, an improvement of 64.5 percentage points. FlowKV also shows very significant performance improvements in Qwen-2.5-7B-Instruct. This indicates that the information flow preserved by FlowKV can effectively enhance the ability to maintain human preferences in multi-turn dialogues.

To further investigate the performance of FlowKV under varying compression strengths and its compatibility with different KV cache management methods, we plotted the user preference following rate as a function of the compression ratio. As shown in Figure 5, we present results for LLaMA-3.1-8B-Instruct with SnapKV and ExpectedAttention, both with and without FlowKV, on the PrefEval benchmark. First, by comparing the full KV cache and the baselines without FlowKV (lines without markers), we observe a substantial performance drop even at low compression ratios.

This finding highlights that user preference following in multi-turn conversations is highly sensitive to the compression ratio. When FlowKV is incorporated, the performance improves markedly. At lower compression ratios (0.1–0.4), FlowKV is able to recover performance to a level close to that of the full KV cache. At higher compression ratios (0.5–0.9), FlowKV continues to provide significant improvements, although the gains are somewhat reduced compared to the low compression ratio. These results demonstrate that FlowKV can robustly preserve information flow across multiple dialogue turns, regardless of the compression ratio. Furthermore, our findings reveal that user preference following in multi-turn dialogue scenarios remains a highly challenging task. Additional

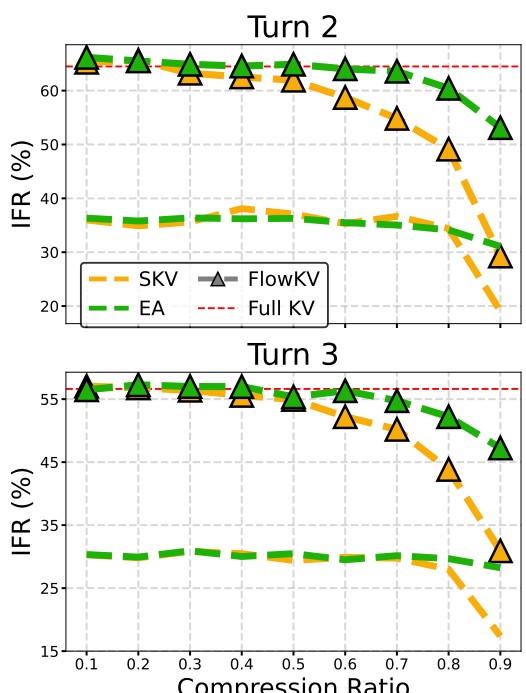

Figure 4: LLaMA-3.1-8B-Instruct on Multi-IF with different compression methods and strategies.

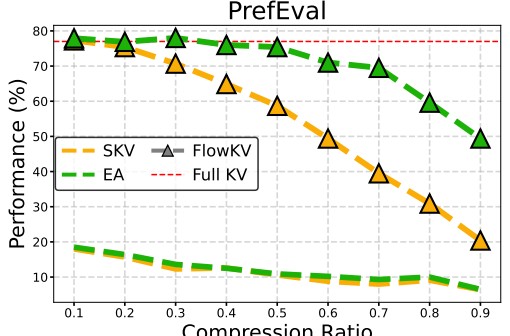

Figure 5: LLaMA-3.1-8B-Instruct on PrefEval with different compression methods and strategies.

Table 3: Efficiency Results for FlowKV

| Configuration | Prompt Length | Output Length | Compression Ratio | Prefilling Time(s) ↓ | Cache Size(GB) ↓ | TTFT (s) ↓ | TPOT (ms) ↓ | Total Gen. Time(s) ↓ |
|---|---|---|---|---|---|---|---|---|
| FullKV | 8192 | 4096 | - | 1.5621 | 1.0000 | 1.6013 | 45.9421 | 184.2934 |
| ChunkKV | 8192 | 4096 | 0.9 | 1.3653 | 0.1000 | 1.3914 | 39.8702 | 164.6600 |
| ChunkKV+FlowKV | 8192 | 4096 | 0.9 | 1.3632 | 0.1000 | 1.4002 | 39.8812 | 165.2100 |

results on long-context multi-turn dialogue, reasoning, and the performance of small models can be found in Appendix B.

## 3.4 EFFICIENCY RESULTS

To evaluate the efficiency of FlowKV, we conduct experiments in a fixed prompt and output length setting, as summarized in Table 3. We compare three configurations: FullKV (baseline without compression), ChunkKV, and ChunkKV integrated with FlowKV. As shown in the table, FlowKV does not compromise the efficiency of existing KV cache compression techniques. It maintains a high compression ratio (0.9) comparable to ChunkKV alone, while introducing negligible overhead in prefilling time and total generation time. Moreover, FlowKV achieves similar performance in terms of TTFT (Time to First Token) and TPOT (Time Per Output Token), demonstrating that it is an efficient strategy that incurs no significant additional computational cost. These results confirm that FlowKV can be seamlessly integrated with KV cache compression methods to further enhance system efficiency without sacrificing speed or resource usage.

## 3.5 LONG-CONTEXT AND REASONING GENERALIZATION

To assess FlowKV's performance on tasks requiring reasoning over long contexts, we conducted new experiments on the Code RepoQA subset of SCBench (Li et al., 2024b). This benchmark is particularly challenging, featuring multi-turn interactions with an average input length of 64K tokens where the model must reason about a code repository. The results, presented in Table 4, show that while the task is inherently difficult (as indicated by the low FullKV scores), our FlowKV mechanism consistently and significantly improves the performance of the base compression method (SnapKV), bringing it much closer to the FullKV baseline. This demonstrates that the benefits of preventing repeated information loss are even more pronounced as dialogue history and context length grow.

Table 4: LLaMA-3.1-8B-Instruct Performance on SCBench (Code RepoQA Subset) with SnapKV

| Method | Compression Ratio | Turn 1 | Turn 2 | Turn 3 | Turn 4 | Turn 5 |
|---|---|---|---|---|---|---|
| FullKV | - | 2.27% | 3.41% | 6.82% | 1.14% | 2.27% |
| SnapKV | 0.1 | 2.27% | 0.00% | 0.00% | 0.00% | 0.00% |
| | +FlowKV | 2.27% | 3.41% | 5.68% | 1.14% | 1.14% |
| SnapKV | 0.3 | 1.14% | 0.00% | 0.00% | 0.00% | 0.00% |
| | +FlowKV | 1.14% | 2.27% | 3.41% | 0.00% | 0.00% |
| SnapKV | 0.5 | 1.14% | 0.00% | 0.00% | 0.00% | 0.00% |
| | +FlowKV | 1.14% | 1.14% | 3.41% | 0.00% | 0.00% |

## 3.6 CASE STUDY

To better illustrate the impact of model generation quality across different multi-turn strategies and KV Cache compression methods, we conduct a case study to demonstrate the performance differences of the LLaMA-3.1-8B-Instruct model when processing multi-turn cumulative instructions on the Multi-IF dataset. Figure 6 shows the Full KV Cache setting successfully follows all instructions of each turn, accurately generating content that satisfies format control, style restrictions, stipulated texts, and other complex tasks. However, the Baseline setting leads to a significant performance crash: not only does content duplication occur in the second turn, violating the newly added "p.p.s" instructions, but in the third turn, the "p.p.s" and double quotation instructions fail to be fully executed, exposing severe context forgetting and the loss of processing capabilities for the conversation flow. When using FlowKV multi-turn conversation, the model shows relative adaptability and performance

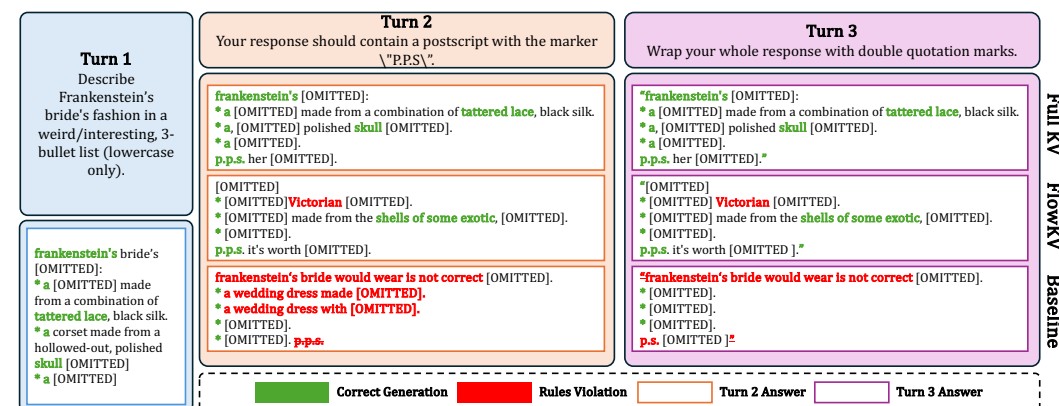

Figure 6: Case Study Comparing Multi-turn Strategies and KV Cache Compression for LLaMA-3.1-8B-Instruct on Multi-IF. Baseline represents SnapKV with 50% compression ratio, and FlowKV represents the FlowKV add on the Baseline. # [OMITTED] indicates portions of the text hidden for brevity.

trade-offs. Although the requirement of all-lowercase format accumulated in the first turn is not fully maintained from the second turn, the content generation still maintains relevance and creativity to the topic and successfully executes the "p.p.s" marks and double quotation marks wrapping instructions newly added in the second and third turns, respectively. This indicates that FlowKV has effectively preserved historical information flow also keep the low KV Cache compression ratio. More case study can be found in Appendix C.

## 4 RELATED WORKS

LLMs recently made significant progress in text understanding and content generation. However, LLMs face challenges when handling multi-turn interactions. Hence, two research fields attract much attention for improving the efficiency and feasibility in real practice: KV Cache optimization on LLMs and multi-turn conversation in LLMs.

### 4.1 KEY–VALUE CACHE OPTIMIZATION STRATEGIES ON LLMS

The attention mechanism needs to compute the relationship between the current token and previous tokens when autoregressive generation. LLMs introduce KV Cache to avoid re-computing the Key and Value vectors of previous tokens. However, the size of KV Cache has a linear relationship with the sequence length (Bai et al., 2023), and it may increase sharply when handling long context or multi-turn conversation. To alleviate the challenges brought by KV Cache, researchers proposed multiple compression and optimization strategies, which can be roughly divided into quantization, sparsification (pruning and eviction), low-rank approximation, and cross-layer sharing. The token sparsification here identifies and selectively retains the KV Cache corresponding to the most important tokens, discarding the relatively unimportant parts. For instance, H2O (Zhang et al., 2023) considers the token with a higher cumulative attention score to be more important, while FastGen (Ge et al., 2023) customizes different strategies for heads by analyzing the behavioral patterns of attention heads. StreamingLLM (Xiao et al., 2023) retains only some of the initial tokens and a fixed-size window of the most recent tokens based on its discovery that these tokens are crucial for maintaining the model's performance. In addition, SnapKV (Li et al., 2024a) compresses KV caches by selecting key attention head features from a prompt, enabling efficient long-sequence processing with minimal performance loss. ChunkKV (Liu et al., 2025b) compresses KV caches by grouping tokens into semantic chunks, preserving contextual meaning while reducing memory usage. Besides, low-rank approximation (Saxena et al., 2024) and cross-layer sharing (Sun et al., 2024; Brandon et al., 2024; Liu et al., 2024) and others are also important optimization directions.

## 4.2 MULTI-TURN CONVERSATION

Multi-turn conversation represents a crucial application scenario for LLMs, simulating human conversation to understand context and generate coherent, relevant responses over successive interaction turns (Zhang et al., 2019; Shuster et al., 2022; Thoppilan et al., 2022; Wu et al., 2020). However, as the number of turns increases, LLMs encounter significant challenges. These include **Long Context Handling**, which involves managing a growing conversation history within the models' limited context windows (Yi et al., 2024); ensuring **Coherence and Consistency**, so that later responses do not contradict earlier parts of the conversation (Li et al., 2025); and maintaining **Long-term Dependency**, which requires models to accurately grasp entities, relationships, and user intentions established early on (Sun et al., 2025). To address these issues, researchers have proposed integrating Retrieval-Augmented Generation (RAG) to improve response accuracy and relevance (Lewis et al., 2020), as well as explicitly modeling the structural information of the conversation to better convey inter-turn dependencies (Yi et al., 2024; Madotto et al., 2019). The evaluation of these capabilities is supported by various benchmarks. For instance, the Multi-IF dataset assesses the continuous adherence to user instructions (He et al., 2024), while PrefEval tests the model's ability to infer, remember, and follow user preferences in long-context conversations (Zhao et al., 2025). Additionally, SCBench serves as a comprehensive benchmark in which multi-turn conversation is a main subtask (Li et al., 2024b).

## 5 CONCLUSION

In this paper, we addressed the critical challenge of KV Cache management in LLMs for multi-turn conversational scenarios, where the trade-off between computational efficiency and contextual coherence is paramount. We identified that conventional KV Cache eviction strategies often lead to severe information degradation due to the repeated compression of historical context. To overcome this limitation, we introduced **FlowKV**, a novel multi-turn isolation mechanism. FlowKV's core principle is to safeguard the integrity of the already compressed KV cache from prior turns, strategically applying compression only to the newly generated segments of the most recent turn. This training-free approach, compatible with existing KV compression techniques, effectively mitigates catastrophic forgetting and preserves vital long-range dependencies.

## ETHICS STATEMENT

This research aims to improve the computational efficiency of Large Language Models in multi-turn conversational settings. Our proposed method, FlowKV, reduces memory and energy consumption, making advanced conversational AI more sustainable and accessible. By enhancing conversational coherence and reducing context forgetting, our work can lead to more reliable and effective dialogue systems. We used public, non-sensitive academic datasets for all experiments and foresee no direct negative ethical implications from this work. We advocate for the responsible deployment of all AI technologies, including more efficient ones.

## REPRODUCIBILITY STATEMENT

To ensure the reproducibility of our findings, we will release the code implementing our FlowKV mechanism and all experimental scripts. Our evaluations were conducted on publicly available models, including LLaMA-3.1-8B and Qwen-2.5-7B, using standard benchmarks such as Multi-IF and PrefEval. The implementation is based on the open-source `kvpress` library, and detailed configurations for all experiments are provided in the appendix.

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

APPENDIX

## A  USE OF LLMs IN PAPER WRITING

We used LLMs solely to aid and polish the writing (e.g., wording refinement and grammar), without generating or altering experimental designs, methods, results, or conclusions. All technical content, analyses, figures, and tables were authored and verified by the researchers.

## B  ADDITIONAL EXPERIMENTAL RESULTS

For this section, we conduct additional experiments to further validate the generalization, robustness, and compatibility of FlowKV.

### B.1  PERFORMANCE WITH SMALLER MODELS

A key point raised was whether FlowKV's performance gains are tied to the strength of the base model. To investigate this, we conducted new experiments with a smaller model, Qwen2.5-1.5B-Instruct. The results, presented in Table 5 for Multi-IF and Table 6 for PrefEval, show that while the absolute performance is lower than with an 8B model (as expected), FlowKV still provides a substantial relative improvement over the baseline compressor. This aligns with FlowKV's design motivation: to mitigate the performance degradation caused by repeated compression in multi-turn dialogues, regardless of the base model's strength.

Table 5: Performance on Multi-IF with Qwen2.5-1.5B-Instruct and SnapKV

| Method | Compression Ratio | IFR R1 | IFR R2 | IFR R3 |
|---|---|---|---|---|
| FullKV | - | 42.02% | 30.18% | 26.23% |
| SnapKV | 0.1 | 44.13% | 14.21% | 12.45% |
| +FlowKV | 44.13% | 29.80% | 26.23% |
| SnapKV | 0.3 | 43.82% | 14.35% | 12.88% |
| +FlowKV | 43.82% | 30.19% | 25.79% |
| SnapKV | 0.5 | 44.71% | 14.46% | 12.72% |
| +FlowKV | 44.71% | 28.55% | 25.94% |

Table 6: Performance on PrefEval with Qwen2.5-1.5B-Instruct and SnapKV

| Method | Compression Ratio | ACC |
|---|---|---|
| FullKV | - | 20.00% |
| SnapKV | 0.3 | 2.20% |
| +FlowKV | 18.70% |
| SnapKV | 0.5 | 1.90% |
| +FlowKV | 15.20% |

### B.2  COMPATIBILITY WITH TRAINABLE COMPRESSION METHODS

FlowKV is a universal, training-free mechanism designed to be orthogonal to the underlying compression algorithm. To demonstrate its compatibility with learnable methods, we conducted an experiment with Activation Beacon (Zhang et al., 2024), a trainable KV Cache compression method. We loaded the pre-trained Beacon-Qwen-2-7B-Instruct model and evaluated it on a subset of 200 samples from the Multi-IF benchmark.

The results in Table 7 show that even with a strong, learnable baseline like Activation Beacon, FlowKV provides significant improvements, particularly in the second turn (R2). This confirms that FlowKV can be effectively combined with trainable compression methods to further mitigate information loss in multi-turn scenarios.

Table 7: Performance on Multi-IF with trainable method (Activation Beacon on Qwen-2-7B-Instruct)

| Method | Compression Ratio | IFR R1 | IFR R2 | IFR R3 |
|--------|-------------------|--------|--------|--------|
| FullKV | - | 49.01% | 26.40% | 10.43% |
| Baseline | 0.1 | 41.38% | 25.46% | 16.01% |
| +FlowKV | | 41.38% | 32.15% | 12.95% |
| Baseline | 0.5 | 42.66% | 25.49% | 16.76% |
| +FlowKV | | 42.66% | 34.11% | 18.49% |

### B.3 DETAILED MULTI-TURN BENCHMARK RESULTS

Figure 7 and 8 provides a detailed turn-by-turn breakdown of the Instruction Following Rate (IFR) for LLaMA-3.1-8B-Instruct and Qwen-2.5-7B-Instruct on the Multi-IF dataset. It compares the performance of four KV cache compression methods (SnapKV, StreamingLLM, ExpectedAttention, and ChunkKV) when integrated with either the FlowKV strategy or a baseline approach. The subplots distinctly illustrate performance at Turn 2 and Turn 3 across a spectrum of compression ratios, with the Full KV Cache performance serving as a reference (red dashed line). This visualization highlights how different compression methods interact with FlowKV and the baseline strategy as conversational depth increases and compression becomes more aggressive.

Figure 9 illustrates the User Preference Following Rate for LLaMA-3.1-8B-Instruct on the PrefEval dataset. The figure contrasts the FlowKV strategy against the baseline strategy across various compression ratios, for different underlying KV cache compression methods (SnapKV, StreamingLLM, ExpectedAttention, and ChunkKV). The performance of a Full KV Cache (uncompressed) is shown as a dashed red line reference. This visualization allows for an assessment of how well user preferences are maintained under increasing compression levels when FlowKV's isolation mechanism is employed compared to a standard baseline approach.

The following table 8, table 10, and table 9 provide comprehensive numerical results complementing the figures. They detail the performance of both LLaMA-3.1-8B-Instruct and Qwen-2.5-7B-Instruct models on the Multi-IF and PrefEval datasets. For the Multi-IF dataset, Instruction Following Rates (IFR) are presented turn-by-turn (Turn 1, Turn 2, and Turn 3). For the PrefEval dataset, the User Preference Following Rate is reported. All results are shown for different KV cache compression methods (SnapKV, StreamingLLM, ExpectedAttention, ChunkKV), comparing the FlowKV strategy against the baseline approach, across a fine-grained spectrum of compression ratios ranging from 0.1 to 0.9. The performance with a Full KV Cache is also provided as a key reference point for each model and turn where applicable.

## C CASE STUDY

In addition, we provide another case study for the PrefEval dataset as Figure 10. Different from Multi-IF, PrefEval requires the model to infer the user preference according to the provided historical conversation, which brings a greater challenge. The case study is conducted on Full KV Cache, 50% SnapKV FlowKV multi-turn conversation, and 50% SnapKV Baseline multi-turn conversation. The user's preference of preferring peer-to-peer interaction is hidden in the previous conversations; for example, the user mentions she prefers learning in group settings. The final query is to make the model recommend resources for data analysis learning. By successfully inferring the user preference, Full KV strategy and FlowKV provide some group-based learning resources. Moreover, the resources provided by Full KV are more authentic. This indicates that FlowKV has effectively preserved historical information but still suffers from some performance losses. However, the Baseline, which is 50% SnapKV Baseline multi-turn conversation here, completely failed to infer the hidden user preferences. Instead, it tended to answer the user's first question from the conversation history. This indicates that in long-context situations, the impact of KV compression is devastating, potentially leading to the generation of unknown responses. Because the conversation history of PrefEval

| KV Method | Strategy | 0.1 | 0.2 | 0.3 | 0.4 | 0.5 | 0.6 | 0.7 | 0.8 | 0.9 |
|---|---|---|---|---|---|---|---|---|---|---|
| | | | | | Turn 1 | | | | | |
| | | | | LLaMA-3.1-8B-Instruct Full KV: 73.41% | | | | | | |
| SKV | Baseline | 74.85% | 76.53% | 74.81% | 73.77% | 76.15% | 75.97% | 74.06% | 75.57% | 76.98% |
| | FlowKV | 74.85% | 76.53% | 74.81% | 73.77% | 76.15% | 75.97% | 74.06% | 75.57% | 76.98% |
| SLM | Baseline | 73.17% | 75.59% | 75.31% | 75.07% | 72.78% | 72.86% | 73.29% | 73.41% | 76.39% |
| | FlowKV | 73.17% | 75.59% | 75.31% | 75.07% | 72.78% | 72.86% | 73.29% | 73.41% | 76.39% |
| EA | Baseline | 74.85% | 75.00% | 75.04% | 74.31% | 76.05% | 75.57% | 75.23% | 75.62% | 76.35% |
| | FlowKV | 74.85% | 75.00% | 75.04% | 74.31% | 76.05% | 75.57% | 75.23% | 75.62% | 76.35% |
| CKV | Baseline | 74.80% | 74.80% | 74.80% | 70.47% | 70.47% | 70.47% | 70.49% | 70.49% | 70.50% |
| | FlowKV | 74.80% | 74.80% | 74.80% | 70.47% | 70.47% | 70.47% | 70.49% | 70.49% | 70.50% |
| | | | | | Turn 2 | | | | | |
| | | | | LLaMA-3.1-8B-Instruct Full KV: 64.49% | | | | | | |
| SKV | Baseline | 35.99% | 34.89% | 35.62% | 38.09% | 37.08% | 35.30% | 36.65% | 34.39% | 19.09% |
| | FlowKV | 65.26% | 65.62% | 63.23% | 62.54% | 61.93% | 58.72% | 54.84% | 49.13% | 29.28% |
| SLM | Baseline | 35.21% | 34.96% | 36.44% | 36.99% | 33.94% | 33.29% | 33.46% | 32.32% | 31.09% |
| | FlowKV | 64.44% | 56.74% | 48.42% | 41.96% | 39.06% | 35.20% | 29.77% | 27.04% | 25.35% |
| EA | Baseline | 36.33% | 35.79% | 36.37% | 36.20% | 36.28% | 35.49% | 35.04% | 34.14% | 31.08% |
| | FlowKV | 66.17% | 65.51% | 64.88% | 64.56% | 64.89% | 64.05% | 63.59% | 60.42% | 53.12% |
| CKV | Baseline | 35.72% | 35.68% | 35.74% | 12.54% | 12.56% | 12.58% | 12.56% | 12.54% | 12.55% |
| | FlowKV | 64.82% | 63.54% | 61.49% | 57.82% | 52.83% | 45.49% | 38.06% | 30.23% | 24.01% |
| | | | | | Turn 3 | | | | | |
| | | | | LLaMA-3.1-8B-Instruct Full KV: 56.62% | | | | | | |
| SKV | Baseline | 30.30% | 29.81% | 30.84% | 30.47% | 29.39% | 29.89% | 29.76% | 28.00% | 17.40% |
| | FlowKV | 57.06% | 56.78% | 56.41% | 55.59% | 54.95% | 52.21% | 50.20% | 43.84% | 30.92% |
| SLM | Baseline | 29.82% | 30.43% | 31.16% | 31.52% | 28.94% | 27.50% | 28.11% | 28.12% | 28.26% |
| | FlowKV | 55.66% | 54.40% | 50.78% | 46.65% | 41.58% | 37.19% | 31.60% | 26.28% | 23.54% |
| EA | Baseline | 30.33% | 29.93% | 30.93% | 30.02% | 30.48% | 29.50% | 30.16% | 29.68% | 28.24% |
| | FlowKV | 56.48% | 57.28% | 57.00% | 56.99% | 55.36% | 56.33% | 54.75% | 52.21% | 47.24% |
| CKV | Baseline | 30.27% | 30.28% | 35.77% | 16.47% | 16.49% | 16.47% | 16.49% | 16.50% | 16.47% |
| | FlowKV | 56.71% | 55.80% | 55.58% | 53.52% | 50.15% | 44.77% | 38.11% | 29.55% | 23.05% |

Table 8: LLaMA-3.1-8B-Instruct Performance on Multi-IF across Different KV Cache Methods and Ratios

| KV Method | Strategy | 0.1 | 0.2 | 0.3 | 0.4 | 0.5 | 0.6 | 0.7 | 0.8 | 0.9 |
|---|---|---|---|---|---|---|---|---|---|---|
| | | | | | Turn 1 | | | | | |
| | | | | Qwen-2.5-7B-Instruct Full KV: 76.30% | | | | | | |
| SKV | Baseline | 77.52% | 76.96% | 76.36% | 75.77% | 76.49% | 72.62% | 50.78% | 49.23% | 71.82% |
| | FlowKV | 77.52% | 76.96% | 76.36% | 75.77% | 76.49% | 72.62% | 50.78% | 49.23% | 71.82% |
| SLM | Baseline | 77.20% | 77.12% | 76.55% | 76.70% | 76.47% | 77.15% | 76.36% | 73.59% | 72.75% |
| | FlowKV | 77.20% | 77.12% | 76.55% | 76.70% | 76.47% | 77.15% | 76.36% | 73.59% | 72.75% |
| EA | Baseline | 76.39% | 77.05% | 77.59% | 77.24% | 75.52% | 75.05% | 75.02% | 73.64% | 72.85% |
| | FlowKV | 76.39% | 77.05% | 77.59% | 77.24% | 75.52% | 75.05% | 75.02% | 73.64% | 72.85% |
| CKV | Baseline | 73.14% | 73.28% | 73.19% | 73.19% | 73.19% | 73.28% | 73.28% | 73.23% | 73.23% |
| | FlowKV | 73.14% | 73.28% | 73.19% | 73.19% | 73.19% | 73.28% | 73.28% | 73.23% | 73.23% |
| | | | | | Turn 2 | | | | | |
| | | | | Qwen-2.5-7B-Instruct Full KV: 60.72% | | | | | | |
| SKV | Baseline | 16.70% | 15.93% | 17.59% | 17.35% | 17.33% | 18.29% | 17.24% | 17.20% | 20.23% |
| | FlowKV | 61.24% | 60.17% | 58.95% | 58.73% | 56.72% | 53.59% | 45.50% | 44.48% | 36.81% |
| SLM | Baseline | 16.77% | 16.39% | 17.23% | 17.13% | 17.31% | 17.66% | 18.66% | 17.77% | 19.35% |
| | FlowKV | 58.05% | 51.67% | 45.64% | 42.63% | 36.82% | 33.83% | 30.31% | 28.46% | 26.48% |
| EA | Baseline | 16.42% | 16.57% | 17.39% | 17.14% | 17.62% | 17.99% | 17.72% | 17.80% | 19.34% |
| | FlowKV | 59.54% | 60.46% | 57.48% | 54.49% | 50.62% | 47.08% | 41.26% | 36.65% | 31.26% |
| CKV | Baseline | 18.47% | 18.46% | 18.47% | 18.46% | 18.47% | 18.47% | 18.48% | 18.48% | 18.47% |
| | FlowKV | 59.52% | 57.18% | 53.82% | 50.99% | 47.27% | 42.68% | 37.25% | 29.78% | 25.03% |
| | | | | | Turn 3 | | | | | |
| | | | | Qwen-2.5-7B-Instruct Full KV: 51.19% | | | | | | |
| SKV | Baseline | 21.27% | 20.77% | 21.83% | 21.77% | 21.96% | 21.84% | 18.83% | 17.98% | 22.35% |
| | FlowKV | 51.33% | 50.97% | 49.41% | 50.07% | 49.67% | 47.11% | 41.61% | 39.74% | 32.41% |
| SLM | Baseline | 21.03% | 20.57% | 20.86% | 20.80% | 21.08% | 21.86% | 22.06% | 22.02% | 22.90% |
| | FlowKV | 47.86% | 44.40% | 42.90% | 39.55% | 35.29% | 32.20% | 30.01% | 26.53% | 22.06% |
| EA | Baseline | 21.77% | 21.16% | 21.94% | 20.93% | 22.00% | 21.34% | 21.78% | 22.06% | 22.90% |
| | FlowKV | 50.84% | 49.74% | 47.11% | 43.69% | 39.25% | 35.22% | 30.02% | 24.89% | 21.75% |
| CKV | Baseline | 21.50% | 21.48% | 21.50% | 21.49% | 21.51% | 21.48% | 21.51% | 21.52% | 21.50% |
| | FlowKV | 49.49% | 49.19% | 47.08% | 46.76% | 43.55% | 40.01% | 35.96% | 29.32% | 22.68% |

Table 9: Qwen-2.5-7B-Instruct Performance on Multi-IF across Different KV Cache Methods and Ratios

| KV Method | Strategy | 0.1 | 0.2 | 0.3 | 0.4 | 0.5 | 0.6 | 0.7 | 0.8 | 0.9 |
|---|---|---|---|---|---|---|---|---|---|---|
| | | | | LLaMA-3.1-8B-Instruct Full KV: 77.00% | | | | | | |
| SKV | Baseline | 18.00% | 15.70% | 12.30% | 12.60% | 10.60% | 8.80% | 8.00% | 9.10% | 6.40% |
| | FlowKV | 77.30% | 75.40% | 70.70% | 64.90% | 58.70% | 49.40% | 39.50% | 30.90% | 20.40% |
| SLM | Baseline | 17.90% | 16.20% | 12.90% | 11.40% | 9.80% | 7.80% | 7.60% | 6.30% | 6.70% |
| | FlowKV | 76.20% | 67.00% | 39.30% | 29.50% | 24.40% | 21.40% | 19.20% | 18.10% | 12.70% |
| EA | Baseline | 18.50% | 16.40% | 13.60% | 12.50% | 10.90% | 10.20% | 9.30% | 10.00% | 6.50% |
| | FlowKV | 77.90% | 76.90% | 78.00% | 76.00% | 75.40% | 71.00% | 69.50% | 59.60% | 49.40% |
| CKV | Baseline | 12.60% | 13.10% | 8.40% | 8.40% | 6.70% | 6.40% | 6.70% | 6.40% | 6.20% |
| | FlowKV | 76.30% | 71.80% | 62.10% | 49.90% | 38.80% | 29.10% | 22.90% | 18.50% | 13.70% |

Table 10: LLaMA-3.1-8B-Instruct Performance on PrefEval across Different KV Cache Methods and Ratios

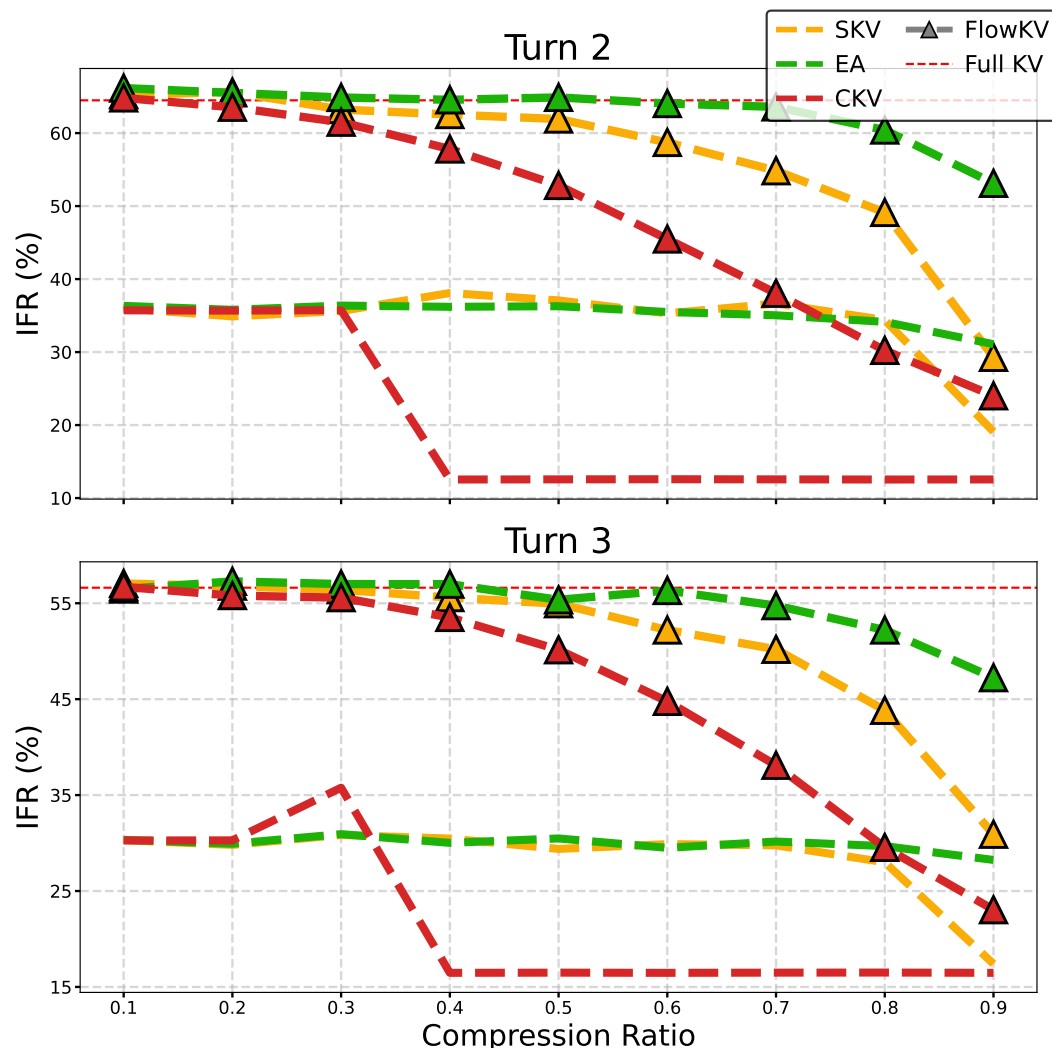

Figure 7: LLaMA3.1-8B-Instruct Multi-IF Instruction Following Performance.

is significantly longer than that of Multi-IF, the performance decline of Baseline will be more pronounced compared to Multi-IF.

# D  THEORETICAL ANALYSIS OF INFORMATION LOSS: THE VANISHING SUPPORT PROBLEM

Existing KV cache compression methods (e.g., SnapKV, H2O, StreamingLLM) primarily operate as *selection operators*, retaining a subset of tokens based on attention scores or importance metrics. We provide a formal analysis using Set Theory to demonstrate that traditional nested compression strategies fundamentally suffer from a "Vanishing Support" problem due to query drift in multi-turn dialogues, a failure mode that FlowKV effectively eliminates.

## D.1  PROBLEM FORMULATION

Let $\mathcal{P} = \{k_1, k_2, \ldots, k_N\}$ be the set of Key-Value (KV) pairs corresponding to the initial context (e.g., the system prompt). For any conversational turn $t$, let $Q_t$ denote the user query. We define the *Oracle Relevance Set* $\mathcal{R}(Q_t) \subseteq \mathcal{P}$ as the subset of prompt tokens strictly required to answer $Q_t$ correctly.

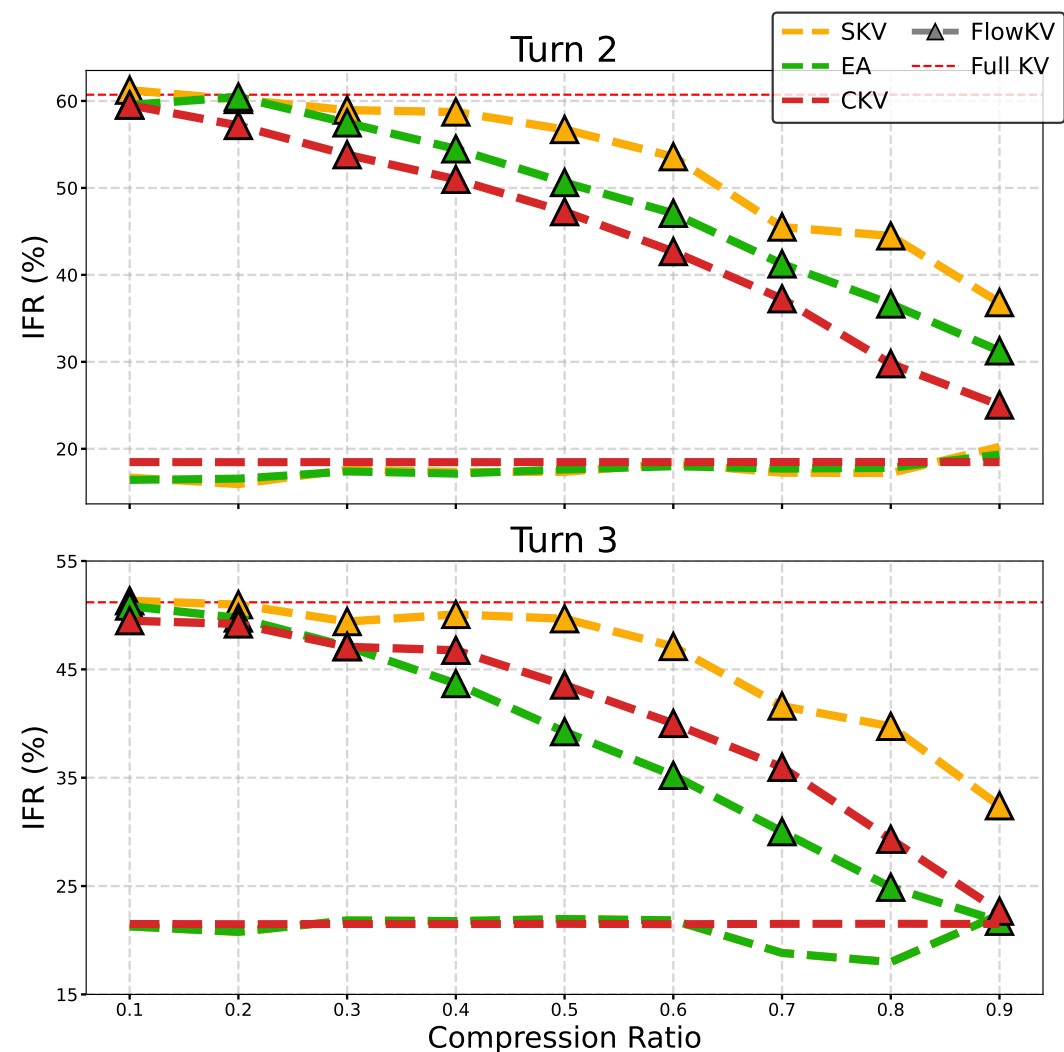

Figure 8: Qwen-2.5-7B-Instruct Multi-IF Instruction Following Performance.

We define a compression operator $\Phi(\cdot \mid Q_t)$ with a budget constraint $\beta$ (where $0 < \beta < 1$), which selects a subset of tokens from the input set based on relevance to $Q_t$.

### D.2 ANALYSIS OF BASELINE: RECURSIVE INTERSECTION

In standard multi-turn strategies, the cache state $C_t$ is derived by compressing the *output of the previous turn* $C_{t-1}$. The retained prompt information at turn $t$, denoted as $C_t^{\text{prompt}}$, follows a recursive definition:

$$C_t^{\text{prompt}} = \Phi(C_{t-1}^{\text{prompt}} \mid Q_t) \qquad (14)$$

Since the compression is selective, $C_t^{\text{prompt}} \subseteq C_{t-1}^{\text{prompt}}$. Consequently, the set of retained tokens at turn $T$ is mathematically the **intersection** of the retained sets across all previous turns:

$$C_T^{\text{prompt}} \subseteq \Phi(\mathcal{P} \mid Q_1) \cap \Phi(\mathcal{P} \mid Q_2) \cap \cdots \cap \Phi(\mathcal{P} \mid Q_T) \qquad (15)$$

This formulation reveals a critical flaw: **Query Drift**. In multi-turn conversations, user queries often shift focus to different aspects of the initial context (e.g., $Q_1$ asks about the first paragraph, $Q_2$ asks about the conclusion). If $Q_i$ and $Q_j$ require disjoint subsets of the prompt (i.e., $\mathcal{R}(Q_i) \cap \mathcal{R}(Q_j) \approx \emptyset$), the recursive intersection forces the support set $C_T^{\text{prompt}}$ to approach the empty set $\emptyset$. Once a token is evicted in turn $t$, it is permanently lost for all turns $k > t$, leading to catastrophic forgetting.

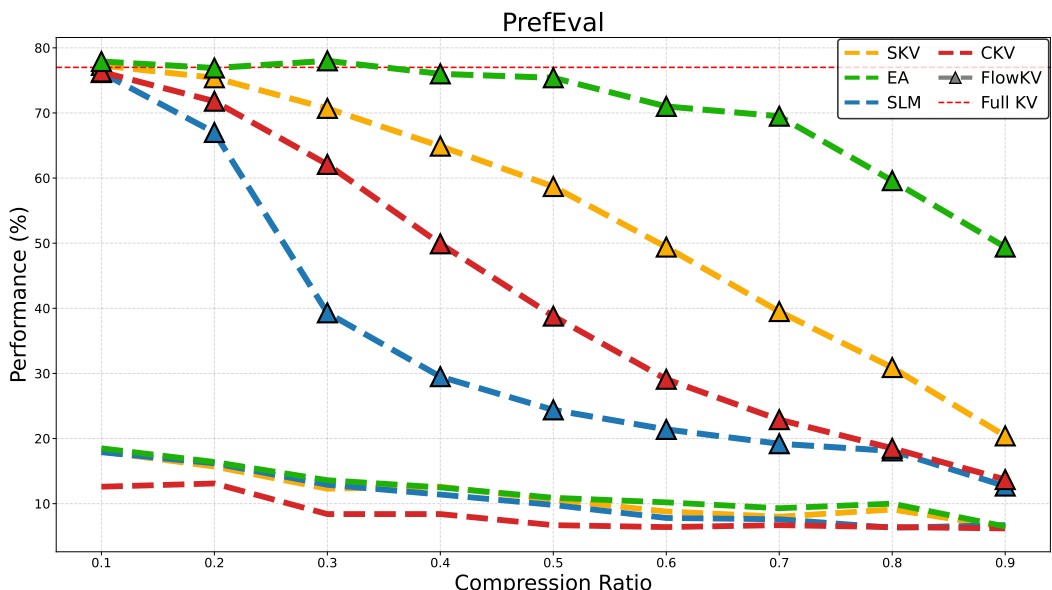

Figure 9: Performance Comparison of Multi-turn Conversation Strategies on LLaMA-3.1-8B-Instruct Using PrefEval Dataset.

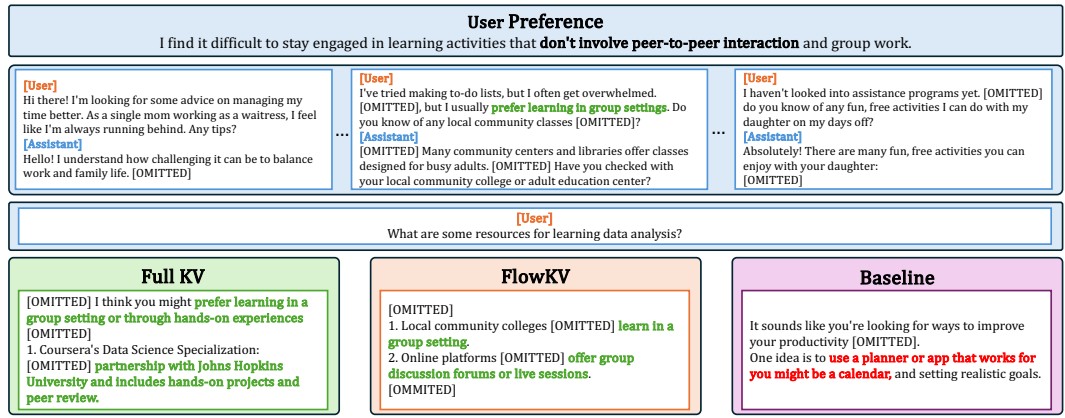

Figure 10: Case Study Comparing Multi-Turn Strategies and KV Cache Compression for LLaMA-3.1-8B-Instruct on PrefEval. Full KV Cache represents no compression, FlowKV represents the FlowKV multi-turn strategy with 50% SnapKV, and Baseline represents the Baseline multi-turn strategy with 50% SnapKV. User preference is hidden in the historical conversation. The model should respond to the final query according to inferred user preference.

## D.3 ANALYSIS OF FLOWKV: PIECEWISE UNION

FlowKV introduces an isolation mechanism. The prompt is compressed once (e.g., based on $Q_1$ or initial attention sinks), yielding a preserved set $S_{\text{frozen}} = \Phi(\mathcal{P} \mid Q_1)$. Crucially, FlowKV enforces that this set remains invariant for all future turns. The effective context available at turn $T$ is constructed via a **union** operation of isolated segments:

$$C_T^{\text{FlowKV}} = S_{\text{frozen}} \cup \Phi(\text{Turn}_1 \mid Q_2) \cup \cdots \cup \Phi(\text{Turn}_{T-1} \mid Q_T) \qquad (16)$$

Regarding the initial prompt information, the support set at turn $T$ satisfies:

$$C_T^{\text{prompt}} = S_{\text{frozen}} = \Phi(\mathcal{P} \mid Q_1) \qquad (17)$$

This implies that the information loss is strictly bounded by the initial compression error and is **independent** of the number of turns $T$ or the drift in subsequent queries $Q_{t>1}$.

### D.4 Error Propagation Bound

We can further quantify this using an error norm $\|\epsilon\|$. Let $H_0$ be the full information of the prompt.

- **Baseline (Exponential Decay):** Assuming an information retention rate $\sigma < 1$ per compression step, the recursive process yields a signal strength of roughly $\|\text{Signal}\| \propto \sigma^T \|H_0\|$.
- **FlowKV (Constant Bound):** The isolation mechanism ensures the signal strength remains $\|\text{Signal}\| \propto \sigma \|H_0\|$, regardless of $T$.

**Conclusion:** The baseline strategy leads to monotonic non-increasing support ($\subseteq$), inevitably causing information collapse as $T \to \infty$. FlowKV maintains a constant lower bound on support, theoretically guaranteeing the preservation of long-term dependencies established in the initial context.

## E Dynamic Budget Allocation for Strict Memory Fairness

A critical aspect of our experimental design is ensuring a fair comparison between FlowKV and baseline methods. This requires that at the end of any given turn, the total size of the KV cache managed by FlowKV is identical to the cache size of a baseline method using the same global compression ratio. We achieve this through a dynamic budget allocation mechanism, which we formalize below.

### E.1 Step-by-Step Algorithm

At each conversational turn $t$, the per-turn compression rate is determined as follows:

1: **Define Target Global Budget:** Calculate the total target cache size for the current turn, $S_{target}(t)$, based on the size the full, uncompressed KV cache would have up to this turn, $S_{full}(t)$, and the fixed global compression ratio, $R_{global}$.

$$S_{target}(t) = S_{full}(t) \times R_{global}$$

2: **Calculate Available Budget for New Data:** Determine the budget available for the new KV pairs generated in the current turn ($Q_t$ and $R_t$), $B_{new}(t)$. This is done by subtracting the size of the already preserved cache from previous turns, $S_{preserved}(t-1)$, from the total target budget.

$$B_{new}(t) = S_{target}(t) - S_{preserved}(t-1)$$

3: **Determine and Apply Local Compression:** The newly generated KV pairs for turn $t$, which have an uncompressed size of $S_{new\_full}(t)$, are compressed to fit precisely into $B_{new}(t)$. This defines the local compression ratio for this specific turn, $R_{local}(t)$.

$$R_{local}(t) = \frac{B_{new}(t)}{S_{new\_full}(t)}$$

This local ratio is then applied to the new KV pairs using the underlying compression algorithm (e.g., SnapKV).

4: **Update Preserved Cache:** The resulting compressed KV pairs from this turn are appended to $S_{preserved}(t-1)$ to create the new, larger preserved cache, $S_{preserved}(t)$, that is carried to the next turn.

This dynamic, per-turn adjustment of the local compression rate ensures we strictly adhere to the global budget while isolating and preserving the integrity of previously compressed history.

### E.2 Discussion on Aggressive Compression in Later Turns

A perceptive observation is that this dynamic approach may require more aggressive compression in later turns (i.e., $R_{local}(t)$ can be higher than $R_{global}$). This is a correct and integral aspect of FlowKV's design.

This constitutes the core design trade-off of FlowKV: we trade potentially stronger, single-pass compression on new information for the perfect fidelity of historical information (by avoiding the cumulative information loss from re-compression). Our experimental results strongly demonstrate that

this trade-off is extremely beneficial. For multi-turn conversations, avoiding catastrophic forgetting of early context (such as system prompts, user identity, and key facts) is far more important than minor variations in the local compression rate of later turns. By isolating historical information, FlowKV fundamentally solves the problem of information decay that plagues traditional methods as the number of turns accumulates.

## F   EVALUATION BENCHMARK

### F.1   DATASET DETAILS

Detailed statistics for each benchmark dataset are provided in Table 11.

| DATASET | TASK TYPE | # TEST | METRIC |
|---|---|---|---|
| Multi-IF (He et al., 2024) | Multi-Turn Instruction Following | 909 | Instruction Following Rate (%) |
| PrefEval (Zhao et al., 2025) | Multi-Turn Preference Inference | 1000 | Preference Following Rate (%) |

Table 11: The statistics of the datasets used in this paper.# TEST denotes the number of test data.

## G   ALGORITHM IMPLEMENTATION DETAILS

To address concerns regarding implementation precision and memory usage, we provide the pseudocode for the FlowKV mechanism. Algorithm 1 explicitly demonstrates two key properties:

1. **Strict Budget Adherence:** The memory usage is strictly bounded by the global compression ratio. We dynamically calculate the budget for the new turn ($B_{new}$) by subtracting the size of the preserved history from the global target size.

2. **Isolation Strategy:** The `Frozen_History` is passed as-is and is never re-compressed. The base compression function $\Phi$ (e.g., SnapKV) is applied *only* to the newly generated KV pairs.

**Algorithm 1** FlowKV Dynamic Budgeting & Isolation Step

**Require:**
 1: $X_{new}$: New input tokens for the current turn (Query).
 2: $\mathcal{H}_{frozen}$: Preserved KV Cache from previous turns (Frozen).
 3: $R_{global}$: Global compression ratio (e.g., 0.5).
 4: $\Phi(\cdot, b)$: Base compression operator (e.g., SnapKV) with budget $b$.
**Ensure:** Updated KV Cache $\mathcal{H}_{updated}$
 5: **// Step 1: Calculate Virtual Context Length**
 6: $\qquad\qquad\qquad\qquad\qquad$ ▷ Total length as if no compression happened (sum of all raw inputs)
 7: $L_{virtual} \leftarrow \mathcal{H}_{frozen}.\text{virtual\_length} + \text{Length}(X_{new})$
 8: **// Step 2: Determine Global Memory Budget**
 9: $\qquad\qquad\qquad\qquad\qquad$ ▷ Strict upper bound on total cache size to ensure fairness
10: $B_{total} \leftarrow \lfloor L_{virtual} \times R_{global} \rfloor$
11: **// Step 3: Calculate Local Budget for New Turn**
12: $\qquad\qquad\qquad\qquad\qquad$ ▷ Remaining space after accounting for frozen history
13: $S_{used} \leftarrow \text{Length}(\mathcal{H}_{frozen}.\text{data})$
14: $B_{new} \leftarrow B_{total} - S_{used}$
15: **if** $B_{new} < \text{Min\_Size}$ **then**
16: $\quad B_{new} \leftarrow \text{Min\_Size}$ $\qquad\qquad\qquad\qquad\qquad$ ▷ Safety floor for extreme cases
17: **end if**
18: **// Step 4: Model Forward & Isolation**
19: $\qquad$ ▷ Generate KV pairs for new tokens. Old history is used for attention but NOT modified.
20: $K_{new}, V_{new} \leftarrow \text{Model.\_forward}(X_{new}, \text{past\_key\_values} = \mathcal{H}_{frozen})$
21: **// Step 5: Compress ONLY New Data**
22: $\qquad\qquad\qquad\qquad\qquad$ ▷ Apply selection/compression only to the current turn's KV pairs
23: $KV_{compressed} \leftarrow \Phi((K_{new}, V_{new}), \text{budget} = B_{new})$
24: **// Step 6: Update History**
25: $\qquad\qquad\qquad\qquad\qquad$ ▷ Append new compressed segment to frozen history
26: $\mathcal{H}_{updated} \leftarrow \text{Concat}(\mathcal{H}_{frozen}, KV_{compressed})$
27: $\mathcal{H}_{updated}.\text{virtual\_length} \leftarrow L_{virtual}$
$\qquad$ **return** $\mathcal{H}_{updated}$

