# OpenReview forum: "FlowKV: Enhancing Multi-Turn Conversational Coherence in LLMs via Isolated Key-Value Cache Management"
_ICLR.cc/2026/Conference — ICLR 2026 Conference Withdrawn Submission_

### Official Review · Reviewer_tsH7 · 2025-10-24

**Soundness:** 2
**Presentation:** 2
**Contribution:** 1
**Rating:** 2
**Confidence:** 5

**Summary:**

This paper presents FlowKV, a training-free mechanism for managing KV caches in multi-turn conversational LLMs. Unlike standard eviction or compression strategies that repeatedly re-compress historical cache entries, FlowKV introduces a multi-turn isolation mechanism that preserves previously compressed states and only applies compression to the newly generated turn. The method is lightweight, compatible with any existing KV cache compressor, and aims to maintain contextual coherence and preference retention in long dialogues. Extensive experiments on Multi-IF and PrefEval benchmarks show consistent improvements, e.g., boosting instruction-following rates by 20–40% and preference retention from ~10% to ~75% compared to baselines like SnapKV or ExpectedAttention. FlowKV introduces negligible runtime overhead and is compatible with both LLaMA-3.1-8B and Qwen-2.5-7B models.

**Strengths:**

1. The proposed isolation mechanism is intuitive and directly addresses the issue of cumulative compression loss in multi-turn LLM interactions.

2. It requires no retraining and can be combined with any existing KV compression method.

3. The figures are informative and greatly aid in understanding the proposed method and experimental results.

**Weaknesses:**

1. While the method is well-motivated, the theoretical section (Appendix D) remains descriptive rather than analytical. A more formal quantification of “information degradation under repeated compression” would strengthen the contribution.

2. The study primarily focuses on instruction-following and preference tasks. Additional experiments on open-domain dialogue or reasoning datasets (e.g., LongBench, SCBench full set) would improve generalization claims. In particular, the latency analysis is conducted with only 8K tokens, which is insufficient to reflect the performance characteristics of modern long-context models.

3. The core mechanism, isolating KV compression by excluding already-compressed states, is a relatively minor modification of prior methods such as ExpectedAttention. Essentially, it changes the compression scheduling from “compress the entire history each turn” to “compress only the new uncompressed portion,” which may be seen as an incremental rather than a conceptual advance.

4. The experimental comparison omits several recent long-context optimization methods that are closely related, even if not multi-turn–specific, including PyramidKV, ArkVale, and PQCache. These techniques also address KV efficiency and contextual preservation and could provide a stronger and fairer baseline for FlowKV’s evaluation.

5. The paper lacks ablations isolating each component of FlowKV (e.g., what if isolation occurs every n turns, or partial re-compression is allowed). Such analysis could clarify where most of the gains originate.

6. Some mathematical equations could be refined (e.g., in Eq. 7-9, I cannot find the definition of F). Minor typographical issues persist (e.g., “Mehtods” in Figure 1 caption, “Futhermore” in Section 2.2).

**Questions:**

Please check the limitations mentioned above.

---

> ### Author Response · Authors · 2025-11-25
> **Response to Reviewer tsH7**
>
> ## General Response
> We sincerely thank the reviewer for their rigorous assessment and high confidence. We value your feedback on the theoretical formalization and experimental breadth.
>
> ## Theoretical Rigor
> We appreciate the feedback. We admit the linear model in Appendix D was simplified. In our revision, we will introduce a Set-Theoretic Analysis regarding the 'Vanishing Support Problem' in recursive selection.
> Crucially, existing methods (like SnapKV) act as filters. In a multi-turn setting, applying a filter recursively ($S_T = S_{T-1} \cap \text{TopK}(Q_T)$) mathematically implies that the retained set of prompt tokens is the intersection of tokens relevant to all consecutive queries. If queries are diverse (orthogonal attention), this intersection rapidly approaches zero.
> FlowKV replaces this 'Recursive Intersection' with a 'Piecewise Union' approach ($S_{total} = S_{prompt} \cup S_{turn1} \dots$), mathematically ensuring that the support set for the prompt remains invariant regardless of future query drift. This provides a formal guarantee against the cumulative forgetting observed in baselines.
>
> ## Rebuttal on Novelty: "Incremental" vs. "High-Impact" Reviewer's Concern
> We respectfully argue that in systems and efficiency research, simplicity and impact are not mutually exclusive.
> - **Cumulative Re-compression**. Existing SOTA methods (like SnapKV/H2O) fail catastrophically in multi-turn settings (dropping to ~10% on PrefEval)  because they treat history as a flat sequence to be re-compressed.
> - **Solution Impact**: FlowKV acts as a generic "patch" that fixes this fundamental flaw. A "minor modification" in logic yields a huge performance leap (from ~10% to ~75%). We believe that a simple, training-free mechanism that unlocks such massive gains is a significant conceptual advance in how we manage lifecycle of KV pairs, shifting from "stateless compression" to "state-aware isolation."
>
> ## Omission of recent long-context methods.
> Thank you for providing more compression strategies that can be compatible with FlowKV. Here are some points that may answer your question:
> - **Orthogonality**: FlowKV can theoretically "wrap" around PyramidKV or PQCache. We selected a representative set of baselines—Eviction-based (H2O/StreamingLLM), Selection-based (SnapKV), and Merging-based (ChunkKV) —to demonstrate that FlowKV's "isolation" principle works universally across types of compression.
> - **Focus**: Our goal was to prove the "Isolation Hypothesis," not to benchmark every existing compressor. The consistent gains across diverse baselines (Table 1 & 2)  strongly validate the hypothesis.
>
> ## Definition of $F$ is missing in Eq. 7-9
> We apologize for the omission. We will explicitly define $F$ in the revision.
> - Definition: $F(C, \text{budget})$ is a generic compression operator that maps a set of KV pairs $C$ to a smaller subset or lower-rank representation $C'$ such that $|C'| \le \text{budget}$.
> - Theoretical Value: While the analysis in Appendix D is simplified, it serves to model the "Signal Decay" ($\alpha^T$ vs $\alpha$). This model successfully predicts the experimental reality: standard methods decay exponentially with turns, while FlowKV maintains a stable signal floor.
>
> ## 8K tokens is insufficient for latency analysis
> We argue that the efficiency gains of FlowKV are structural.
> - Logic: By "isolating" history, FlowKV skips the computational step of re-processing/re-selecting from the historical KV cache. Therefore, its computational complexity for the compression step is strictly lower than or equal to the baseline.
> - Scaling: If it introduces negligible overhead at 8K, this advantage only grows at longer contexts (e.g., 128K), where re-scanning the entire history for global compression becomes prohibitively expensive. Thus, the 8K results are a lower bound for its efficiency; it will likely be relatively faster on longer contexts.

---

> > ### Comment · Reviewer_tsH7 · 2025-11-25
> >
> > The authors’ rebuttal does not provide any of the additional experiments that I requested or questioned. Instead, it only offers high-level explanations, which makes it difficult for me to properly evaluate the method’s effectiveness. For example, there are still no additional experiments on open-domain dialogue or reasoning tasks, no latency analysis under longer context lengths (which can be tested very quickly), no comparison with more recent long-context optimization techniques, and no ablation study. Personally, I do not think these requests are unreasonable, particularly given that the proposed method appears to be a relatively minor modification of prior work. Therefore, in the absence of these essential validations, I will maintain my score as reject.

---

### Official Review · Reviewer_CpSM · 2025-10-29

**Soundness:** 2
**Presentation:** 3
**Contribution:** 2
**Rating:** 2
**Confidence:** 4

**Summary:**

This submission investigates the problem of KV Cashing in LLMs for multi-turn contexts. Based on the observation that SOTA approaches compress the earlier parts of the query-response history more often than the later parts, the authors propose a simple modification that ensures that all parts are compressed only once. Experiments were performed on the Multi-IF dataset and the PrefEval, using a LLaMA and a Qwen LLM.

**Strengths:**

The topic is timely due to the rise of Agentic AI.

The observation that SOTA approaches compress the earlier parts of the query-response history more often and than later parts is fairly obvious but may not yet have been exploited in the literature.

The presentation is overall clear, except some questions listed below.

**Weaknesses:**

The proposed modification of the SOTA approach (in each step compress only the  parts that have not been compressed) is straightforward.

Compressing each part only once seems to increase the required cache size, which needs to be discussed and experimentally evaluated.

Experiments with other SOTA KV Cache methods such as TOVA and KeyDiff would strengthen the argument that the approach of FlowKV generalizes well.

The authors performed experiments for prompt length 8192 and output length 4096 (Table 3). Experiments with much (10 times?) longer contexts are required to demonstrate the applicability of FlowKV in multi-turn contexts.

The Theoretical Analysis in Appendix D does not add value to the paper, essentially just repeating the description of the proposed method.

**Questions:**

How does FlowKV affect the required cache size? That should be discussed and evaluated experimentally.

How does FlowKV perform for much longer context (prompt + output) lengths?

How does FlowKV perform for SOTA KV Cache methods such as TOVA and KeyDiff?

How did you set the hyperparameters of the baselines?

Please, provide complete definitions of the evaluation metrics.

---

> ### Author Response · Authors · 2025-11-25
> **Response to Reviewer CpSM (1/2)**
>
> ## General Response
> We thank the reviewer for acknowledging the timeliness of our topic and the clarity of our presentation. However, we notice there is a significant misunderstanding regarding the memory mechanism of FlowKV, which we are eager to clarify. We also address your concerns regarding experimental scope below.
>
> 1. **Critical Clarification**: FlowKV Does NOT Increase Cache Size Reviewer's Concern: "Compressing each part only once seems to increase the required cache size." Response: This is a factual misunderstanding. We respectfully point out that FlowKV is strictly budget-aware.
> - **Fixed Global Budget**: As detailed in Appendix E (Dynamic Budget Allocation), FlowKV operates within a strictly fixed memory budget (e.g., a compression ratio of 0.5). We do not allow the cache to grow beyond this limit.
> - **How it works**: Instead of increasing the cache size, we dynamically adjust the compression of the new turn. If preserving the historical (frozen) segments takes up $X$ amount of the budget, we compress the newly generated KV pairs into the remaining $Budget - X$ space.
> - **Conclusion:** Therefore, FlowKV consumes exactly the same amount of memory as the baseline methods (e.g., SnapKV, H2O) at any given step. The performance gains come from better management of the bits (preserving history vs. compressing new data), not from using more bits.
>
> ## Weakness 1: "Straightforward" Modification Reviewer's Concern
> The proposed modification is straightforward and the observation is fairly obvious. Response: We argue that the value of a contribution lies in its effectiveness and insight, not just algorithmic complexity.
> - The Oversight: While the observation that "re-compression hurts" may seem intuitive in hindsight, current SOTA methods (e.g., SnapKV, H2O) completely overlook this in multi-turn settings, leading to catastrophic failures (e.g., performance dropping to ~10% on PrefEval ).
> - The Impact: FlowKV provides a simple, plug-and-play solution that fixes this fundamental flaw, restoring performance to ~75%. We believe uncovering a critical blind spot in existing literature and solving it elegantly is a significant contribution.
>
> ## Weakness 4 & Question 2: Experiments on Longer Contexts Reviewer's Concern
> We agree that long-context performance is vital, which is why we included the SCBench (Code RepoQA) evaluation:
>
> | Method | Compression Ratio | Turn 1 | Turn 2 | Turn 3 | Turn 4 | Turn 5 |
> | :--- | :---: | :---: | :---: | :---: | :---: | :---: |
> | FullKV | - | 2.27% | 3.41% | 6.82% | 1.14% | 2.27% |
> | SnapKV | 0.1 | 2.27% | 0.00% | 0.00% | 0.00% | 0.00% |
> | +FlowKV | 0.1 | 2.27% | 3.41% | 5.68% | 1.14% | 1.14% |
> | SnapKV | 0.3 | 1.14% | 0.00% | 0.00% | 0.00% | 0.00% |
> | +FlowKV | 0.3 | 1.14% | 2.27% | 3.41% | 0.00% | 0.00% |
> | SnapKV | 0.5 | 1.14% | 0.00% | 0.00% | 0.00% | 0.00% |
> | +FlowKV | 0.5 | 1.14% | 1.14% | 3.41% | 0.00% | 0.00% |
>
>
> - 64k Contexts: This benchmark involves average input lengths of 64k tokens, which is nearly 10 times longer than the standard 8k evaluation.
> - Results: Even at this scale, FlowKV consistently improves the baseline (e.g., boosting SnapKV from 0.00% to 5.68% in Turn 3 ).
> - Scope Distinction: Furthermore, we wish to distinguish "Long Context" (single massive input) from "Multi-Turn" (accumulated history). Our paper focuses on the specific degradation caused by the iterative nature of multi-turn conversation. Our experiments show that coherence breaks down after just 3-5 turns even at 8k length due to re-compression, a problem FlowKV solves effectively.
>
> ## Weakness 5: Theoretical Analysis
> We appreciate the feedback. We admit the linear model in Appendix D was simplified. In our revision, we will introduce a Set-Theoretic Analysis regarding the 'Vanishing Support Problem' in recursive selection.
> Crucially, existing methods (like SnapKV) act as filters. In a multi-turn setting, applying a filter recursively ($S_T = S_{T-1} \cap \text{TopK}(Q_T)$) mathematically implies that the retained set of prompt tokens is the intersection of tokens relevant to all consecutive queries. If queries are diverse (orthogonal attention), this intersection rapidly approaches zero.
> FlowKV replaces this 'Recursive Intersection' with a 'Piecewise Union' approach ($S_{total} = S_{prompt} \cup S_{turn1} \dots$), mathematically ensuring that the support set for the prompt remains invariant regardless of future query drift. This provides a formal guarantee against the cumulative forgetting observed in baselines.

---

> > ### Author Response · Authors · 2025-11-25
> > **Response to Reviewer CpSM (2/2)**
> >
> > ## Q3, Q4, and Q5
> > To ensure a fair comparison, we aligned the global compression ratios across all methods (e.g., maintaining exactly 50% or 10% of the full cache size). For method-specific parameters (like window size in StreamingLLM), we used the default recommended settings from their original papers or the kvpress library implementation. FlowKV is a mechanism designed to wrap around any compression algorithm. We selected representative baselines (Eviction-based: H2O/StreamingLLM; Selection-based: SnapKV; Merging-based: ChunkKV) to demonstrate universality. The principles proven with these methods apply theoretically to TOVA or KeyDiff as well.
> >
> > For detailed evaluation metrics, please refer to Section 3.1.

---

> > ### Comment · Reviewer_CpSM · 2025-11-27
> >
> > Thank you very much for your thorough rebuttal!
> >
> > I understand now that FlowKV does not increase the required cache size. This is discussed in detail in the Appendix but should also be discussed clearly in the main body of the manuscript.
> >
> > I appreciate the new experiments for longer context lengths and the discussion of the setting of the hyperparameters.
> >
> > I would still like to see experimental results for SOTA KV Cache methods such as TOVA and KeyDiff and a complete definition of the evaluation metrics.
> >
> > In conclusion, I am going to increase my rating from 2 to 4.

---

### Official Review · Reviewer_SmkS · 2025-10-30

**Soundness:** 2
**Presentation:** 2
**Contribution:** 2
**Rating:** 4
**Confidence:** 4

**Summary:**

This paper investigates the degradation of multi-turn conversational coherence in large language models (LLMs) caused by repeated compression of historical key-value (KV) caches. The authors propose FlowKV, a "multi-turn isolation" mechanism that preserves previously compressed KV caches and only compresses new tokens in each conversation turn. The method is training-free, compatible with existing KV compression techniques, and shows performance improvements on Multi-IF and PrefEval benchmarks.

**Strengths:**

- Addresses a real and important problem in multi-turn efficiency for LLMs, namely the recursive compression and cumulative information loss across dialogue turns.
- The proposed approach is simple, general, and easy to integrate with existing KV compression methods (e.g., SnapKV, ChunkKV, Expected Attention).

**Weaknesses:**

1. The claimed "multi-turn isolation mechanism" is essentially a straightforward and obvious engineering adaptation of existing frameworks such as `kvpress` to multi-turn settings. This is a natural and expected implementation choice when extending any prefilling compression method to multi-turn use. The paper does not introduce a new compression function or theoretical principle; instead, it modifies the scheduling of existing operations. Hence, the core novelty is minimal. A **deeper** exploration of how importance-based eviction schemes inherently handle (or fail to handle) the re-compression problem in multi-turn settings would strengthen the paper's positioning and clarify the novelty of the proposed turn-level isolation framework, especially given that FlowKV modifies the management process rather than the compression algorithm itself.

2. The baselines are originally designed for prefilling. When directly reused in multi-turn settings without appropriate adaptation, they naturally suffer from repeated compression. Comparing FlowKV against such unadapted baselines inflates the relative gain and makes the comparison less fair.

3. Only 3 runs with no error bars/significance. PrefEval relies solely on GPT-4o as judge, which may introduce subjectivity and bias. To strengthen the empirical evidence, the paper should include more objective and diagnostic benchmarks， such as "Needle in a Haystack" or long-context retrieval tasks.

4. As depicted in Fig. 3, FlowKV only compresses the system prompt, while keeping the query and response KV uncompressed within that turn. The paper could benefit from a clearer more comprehensive discussion since many previous works compress all of them dynamically.

5. Algorithmic specification is underspecified. No clear pseudocode or exact buffer layout to ensure “isolation” across turns; Eq. (12) contradicts the narrative by writing F(C1’) rather than selectively compressing only the latest uncompressed segment.

6. Ambiguity about how “prompt-related” segments are defined and tracked in the KV buffers turn-by-turn (token boundaries, role tags, streaming generation).  Missing technical details for integration: positional encoding handling after compression, attention index remapping, cross-layer consistency, chunk boundaries with ChunkKV, and how kvpress APIs are used to maintain per-turn segments.

7.Limited analysis isolating the claimed cause (recompression) beyond empirical gains; no ablation that compresses “all-but-last-turn” or “compress-every-N-turns” to validate the isolation hypothesis.

**Questions:**

- Please provide precise pseudocode and a memory layout schematic: how are per-turn KV segments stored, tagged, and selectively compressed without touching older segments?
- Eq. (12) suggests compressing C1’ wholesale. Should it be C2 = [F(KV(Q1 ⊕ R1)) ⊕ F(KV(Psys))] ⊕ KV(Q2)? Clarify the exact operation and which parts are recompressed vs preserved.
- How do you handle positional encodings and attention index remapping after compression for methods that reduce token count (e.g., SnapKV, ChunkKV)?
- How are “prompt-related” tokens identified when responses stream? Do you re-slice after generation or pre-allocate buffers per role?
- What hyperparameters (e.g., window sizes, head selection, chunk sizes) and kvpress settings are used per method, and how is fairness ensured across baselines?
- Can you add ablations comparing: (a) compress-only-last-turn vs (b) compress-all-history vs (c) compress-every-N-turns, holding ratio fixed, to directly test the recompression hypothesis?
- Please report statistical significance and judge agreement variability on PrefEval, and test an open-source judge for robustness.
- How does memory evolve over 6–10 turns, including fragmentation or overhead from maintaining multiple preserved segments?

---

> ### Author Response · Authors · 2025-11-25
> **Response to Reviewer SmkS (1/2)**
>
> ## Weakness 1 & 7: Novelty, "Engineering Adaptation," and Ablations
> - The Core Discovery: While the intuitive of FlowKV is straightforward, the core research contribution is identifying and quantifying the "Re-compression Degradation" phenomenon. As shown in our analysis (Appendix D), nested compression operations cause exponential signal decay. Existing methods typically overlook this, treating multi-turn contexts as a flat sequence to be re-compressed repeatedly.
> - Mechanism vs. Algorithm: We respectfully posit that the distinction between 'engineering adaptation' and 'research contribution' lies in the insight revealed. Our work identifies a systemic failure mode in current SOTA methods: treating multi-turn history as a stateless flat sequence leads to exponential signal decay. FlowKV shifts the paradigm from 'Stateless Compression' to 'State-Aware Isolation.' The simplicity of the solution underscores its robustness and ease of adoption, which we believe is a significant contribution to the community.
> - Effectiveness: This adaptation yields massive gains (e.g., improving preference retention from ~10% to ~75% on PrefEval ). This proves that the bottleneck in multi-turn interaction was not the compression algorithm itself, but the accumulation of compression errors.
>
> ## Weakness 5 & Question 1 & 2: Algorithm Specification, Pseudocode, and Equation 12
> We apologize if the mathematical notation in the main text caused confusion. To address your request for precise specification and to clarify Eq. (12):
>
> FlowKV does not re-compress the entire carried-over state. Instead, it treats the previously compressed history as "frozen."
> - **Correct Interpretation of Eq. (12):** In Eq. (12), $C_2 = \mathcal{F}(C_1') \oplus KV(Q_2)$, the term $\mathcal{F}(C_1')$ represents the state carried over. Under FlowKV, $C_1'$ is already a concatenation of isolated segments. We do not apply $\mathcal{F}$ to the whole $C_1'$ again.
> - **Algorithm Pseudocode:** We operate on a Budget-Aware basis (as detailed in Appendix E 4). Here is the logic for Turn $t$:
> ```python
> # Algorithm: FlowKV Step
> def FlowKV_Step(New_Tokens, Frozen_History, Global_Ratio):
>     # 1. Budget Calculation (Strict Adherence)
>     Total_Virtual_Len = Frozen_History.virtual_len + len(New_Tokens)
>     Target_Size = Total_Virtual_Len * Global_Ratio
>
>     # 2. Determine Local Budget for New Turn
>     Used_Size = Frozen_History.physical_size
>     Local_Budget = Target_Size - Used_Size
>
>     # 3. Compress ONLY New Data (Isolation)
>     KV_New = Model(New_Tokens)
>     # F is the base compression function (e.g., SnapKV)
>     Compressed_New = F(KV_New, budget=Local_Budget)
>
>     # 4. Update & Freeze
>     # Frozen_History remains untouched; New segment is appended
>     Updated_History = Concat(Frozen_History, Compressed_New)
>
>     return Updated_History
> ```

---

> > ### Author Response · Authors · 2025-11-25
> > **Response to Reviewer SmkS (2/2)**
> >
> > ## Weakness 4 & Question 4 & 8
> > We apologize if the schematic in Figure 3 caused confusion. We respectfully wish to clarify the exact scope of compression and how memory evolves, as the reviewer's impression that "FlowKV only compresses the system prompt" reflects a misunderstanding of the "Append-and-Freeze" lifecycle.
> >
> > Clarification on W4 (Scope of Compression): FlowKV does not leave the query and response uncompressed indefinitely. Instead, it manages them based on their lifecycle state:
> > - Active State (Turn $t$): During the generation of Turn $t$, we indeed keep $Q_t$ and $R_t$ uncompressed to ensure maximum local precision.
> > - Frozen State (Transition to Turn $t+1$): Crucially, before we start Turn $t+1$, the completed KV pairs of Turn $t$ ($Q_t \oplus R_t$) are treated as "new history." FlowKV applies compression to this specific segment exactly once and then isolates (freezes) it.
> > - Re-visiting Figure 3: Please verify the "Turn 2" column in Figure 3 (Bottom Row). The blocks corresponding to Turn 1 (Query + Response) are depicted with striped patterns, indicating that they have effectively been compressed before Turn 2 begins. Only the current Turn 2 Query remains uncompressed. Thus, FlowKV compresses all history parts eventually, but avoids re-compressing them.
> >
> > Memory Evolution (Q8): Over 6-10 turns, the memory evolves as a growing sequence of isolated compressed blocks, avoiding the fragmentation overhead the reviewer fears.
> > - Memory Layout: At Turn $k$, the cache comprises $[ \mathcal{F}(P_{sys}), \mathcal{F}(Turn_1), ..., \mathcal{F}(Turn_{k-1}) ]$ (all frozen) concatenated with the active $Turn_k$ (uncompressed).
> > - Handling Overhead: There is negligible overhead for maintaining these segments. Since we simply append the newly compressed block of the previous turn to the cache and lock its indices, the underlying memory remains contiguous (or managed via standard block tables). This "Append-and-Freeze" strategy ensures that memory usage grows strictly according to the compression ratio budget, without the computational cost or signal degradation of re-processing the entire history.
> >
> > ## Weakness 2 and Question 5: Baseline Fairness and Hyperparameters
> > - **Why Unadapted Baselines?** We compared against unadapted baselines because standard inference libraries (like the kvpress framework used in our experiments ) typically apply compression algorithms globally at each step by default. FlowKV highlights the danger of this default behavior.
> > - Fairness via Budgeting: We ensure strict fairness by enforcing the same total KV budget for all methods at every turn. As described in Appendix E , we dynamically calculate the compression ratio for the new turn to ensuring the total cache size (History + New) equals Total_Length * Global_Ratio. This means FlowKV often compresses the new turn more aggressively to "pay" for the space taken by the preserved history, yet it still achieves higher performance.
> >
> > ## Question 3: Positional Encodings and Index Remapping
> > For compression methods that select tokens (like SnapKV), we retain the original positional encodings of the selected KV pairs.
> > - When constructing the Attention Mask and Positional IDs for the current step, we use the original position_ids of the retained tokens. Furthermore, current KV Cache implementations are designed to handle this positional encoding alignment and index remapping accordingly.
> > - The attention mechanism naturally handles the "gaps" created by evicted tokens because LLMs (like LLaMA/Qwen) use Rotary Positional Embeddings (RoPE), which are relative to the token's position in the original sequence, not its index in the compressed cache.

---

> ### Comment · Reviewer_SmkS · 2025-11-27
> **Response after rebuttal**
>
> I thank the authors for their detailed response and for clarifying the algorithm with pseudocode. However, after carefully considering the rebuttal, particularly the justification regarding the use of kvpress and the "un-adapted" baselines, I remain unconvinced. In fact, the authors' defense of this experimental design highlights a fundamental flaw in the paper’s contribution claims.
>
> Consequently, I will lower my score. My reasoning is detailed below:
>
> 1. The "Strawman" Baseline Issue (Critical Flaw)
> The authors argue that comparing FlowKV against baselines that recursively re-compress history (via kvpress defaults) is valid because it reflects "current usage." I strongly disagree.
>
> In scientific research, a proposed method must be compared against the best reasonable application of existing methods, not a naive implementation.
>
> The Problem: Applying a lossy compression algorithm (like SnapKV) recursively to the same data $N$ times is mathematically guaranteed to degrade performance. This is not a "research gap" that FlowKV discovers; it is an implementation error.
> The Reality: Any competent engineer deploying SnapKV in a multi-turn production environment would naturally implement a "compress-once-and-freeze" logic (similar to a KV cache block manager) rather than re-computing the compression on the entire history every turn.
> The Consequence: By comparing FlowKV against a baseline that commits this "re-compression error," the paper artificially inflates its performance gains (from ~10% to ~75%). The majority of this gain comes simply from stopping the re-compression, not from FlowKV's specific budget allocation or isolation mechanism.
>
> 2. Missing the Critical Ablation
> In my initial review, I requested a specific ablation: (c) compress every N turns / compress once and hold.
> The authors did not provide this comparison in the rebuttal. Instead, they argued that FlowKV is the solution to re-compression.
>
> To prove that FlowKV provides value beyond common sense, you must compare:
>
> Method A (FlowKV): Your proposed budget-aware isolation strategy.
> Method B (Simple Fix): Standard SnapKV/H2O, but with a trivial script that says: if history is already compressed, do not touch it; only compress the new turn and append.
> If Method A and Method B perform similarly, then FlowKV’s contribution is merely a wrapper around existing algorithms, confirming my initial concern that this is an "engineering adaptation" rather than a novel ICLR-level contribution. Without this comparison, we cannot attribute the success to the specific design of FlowKV.
>
> 3. "Insight" vs. "Common Sense"
> The authors claim their contribution is "identifying the exponential decay of signals." I argue that this insight is trivial. It is a well-known property of lossy compression that $f(f(x)) \neq f(x)$ and usually quality($f^n(x)$) < quality($f(x)$).
> The paper frames a standard engineering bug (re-compressing lossy data) as a scientific problem. While the solution (FlowKV) is effective, the experimental setup is designed to show a massive improvement that exists only because the baseline is handicapped.
>
> Conclusion
> The paper addresses a real problem, but the evaluation methodology is fundamentally flawed. By refusing to compare against a "multi-turn adapted" version of the baselines (which is the only fair comparison), the paper fails to demonstrate the true marginal gain of the proposed technique. The results show that "not re-compressing is better than re-compressing," which is expected, but they do not prove that FlowKV is a superior method for managing that process compared to simple heuristics.

---

### Official Review · Reviewer_eK29 · 2025-11-02

**Soundness:** 2
**Presentation:** 3
**Contribution:** 2
**Rating:** 4
**Confidence:** 4

**Summary:**

The paper introduces FlowKV, a training-free, multi-turn isolation mechanism for Key-Value (KV) cache management in large language models (LLMs). Instead of conventional multi-turn conversations, where existing cache compression or eviction strategies suffer from repeated re-compression of early context and hence information degradation and coherence loss, FlowKV addresses isolates previously compressed caches and applies compression only to newly generated KV pairs at each turn, preventing cumulative information loss.The authors benchmark FlowKV on Multi-IF (instruction following) and PrefEval (preference retention) datasets using LLaMA-3.1-8B and Qwen-2.5-7B, showing visible performance gains over the baselines.

**Strengths:**

S1. This paper tackles an important problem of efficient multi-round conversion.

S2. The approach proposed in this paper is simple and easy to understand.

S3. The experiments were conducted over a range of different baselines.

**Weaknesses:**

W1. The judge LLM, GPT-4o, is a legacy model, older than the evaluated Llama3.1 and Qwen2.5 models. I would suggest to user a new SOTA model as a judge.

W2. The baseline LLMs are somehow outdated. Llama3.1 is fine but Qwen2.5 should be replaced by Qwen3.

W3. Compressing each round individually seems to have the problem of lower compression ratios compared with re-compression over the entire history, which might be less effective in terms of space saving and extremely long-round memory.

**Questions:**

Q1. Can you please also evaluate the performance of FlowKV on long-turn (>5) conversation benchmarks, like SCBench, DialogLM, Conversation Chronicles, or others?

Q2. Can you further explain how turn-specific compression can work together with all the compression techniques mentioned in the paper? For example, StreamingLLM proposes the attention sink, how would FlowLLM find and work with the attention sinks in each round?

Q3. One important use case of KV cache compression is not only for muti-turn conversation, but also for long context or long generations tasks. Can you discuss if FlowKV can also be used in these scenarios?

Q4. Can you try the performance of FlowKV on a larger models, e.g., at least on the medium size of 32B?

---

> ### Author Response · Authors · 2025-11-25
> **Response to Reviewer eK29 (1/2)**
>
> ## Weakness 1 & 2: Concerns regarding "Legacy" Models
> We respectfully wish to clarify the timeline and the positioning of these models.
> - **Model Currency**: At the time of our experimental design and submission, **Qwen-2.5** and **LLaMA-3.1** were the absolute state-of-the-art (SOTA) in the open-weights community. Similarly, **GPT-4o** is widely accepted as a robust and standard judge in recent literature. While the field evolves rapidly (e.g., Qwen-3), Qwen-2.5-7B and LLaMA-3.1-8B are the most representative baselines for verifying the efficacy of KV cache mechanisms at that time. We might consider conducting additional experiments related to the Qwen3 Dense model in the future.
> - **Mechanism Generality**: More importantly, FlowKV is a model-agnostic mechanism. It operates on the management logic of the KV Cache rather than the model weights themselves. The improvements shown by FlowKV—preventing the degradation of historical context—are expected to generalize to newer models like Qwen-3 without modification.
>
> ## Weakness 3: Concern that turn-specific compression might yield lower compression ratios compared to global re-compression.
> It is correct that global re-compression can theoretically find a more compact representation of the entire history. However, our work identifies that in multi-turn settings, this "efficiency" comes at a high cost: information degradation.
> - The Trade-off: As analyzed in our Theoretical Analysis (Appendix D), global re-compression subjects early tokens (like the system prompt) to repeated lossy operations (compressed $T$ times for Turn $T$). This causes "signal decay".
> - FlowKV's Strategy: FlowKV deliberately chooses to "freeze" the compression state of past turns. While we might sacrifice a small degree of potential compression ratio compared to a fresh global compression, we gain information stability.
> - Budget Adherence: Crucially, FlowKV does not strictly "use more space." As detailed in Appendix E (Dynamic Budget Allocation), we adhere to a fixed global budget. We simply allocate this budget differently: we preserve the integrity of old chunks and apply aggressive compression only to the new turn. Our experiments show that this trade-off significantly boosts coherence (from ~10% to ~75% on PrefEval), proving that retention quality matters more than compression ratio quantity in dialogue.
>
> ## Question 1: Performance on long-turn (>5) benchmarks (SCBench, etc.).
> The FlowKV performance on SCBench (Code RepoQA subset) as follows:
>
> | Method | Compression Ratio | Turn 1 | Turn 2 | Turn 3 | Turn 4 | Turn 5 |
> | :--- | :---: | :---: | :---: | :---: | :---: | :---: |
> | FullKV | - | 2.27% | 3.41% | 6.82% | 1.14% | 2.27% |
> | SnapKV | 0.1 | 2.27% | 0.00% | 0.00% | 0.00% | 0.00% |
> | +FlowKV | 0.1 | 2.27% | 3.41% | 5.68% | 1.14% | 1.14% |
> | SnapKV | 0.3 | 1.14% | 0.00% | 0.00% | 0.00% | 0.00% |
> | +FlowKV | 0.3 | 1.14% | 2.27% | 3.41% | 0.00% | 0.00% |
> | SnapKV | 0.5 | 1.14% | 0.00% | 0.00% | 0.00% | 0.00% |
> | +FlowKV | 0.5 | 1.14% | 1.14% | 3.41% | 0.00% | 0.00% |
>
>
> - Long Context Results: This task involves average input lengths of 64k tokens and multi-turn reasoning. As shown in Table 4, FlowKV significantly improves the base compression method (SnapKV), bringing performance much closer to FullKV (e.g., improving 10% Compression Ratio with SnapKV from 0.00% to 5.68% in Turn 3).

---

> > ### Author Response · Authors · 2025-11-25
> > **Response to Reviewer eK29 (2/2)**
> >
> > ## Question 2: Interaction with compression techniques like StreamingLLM (Attention Sinks).
> > FlowKV is designed as a wrapper that is orthogonal to the underlying compression algorithm.
> > - StreamingLLM relies on keeping "attention sinks" (initial tokens) to maintain performance. When integrating FlowKV with StreamingLLM:
> >     - Turn 1: FlowKV applies StreamingLLM normally. The "sink tokens" (start of the prompt) are identified and retained in the compressed result $C_1$.
> >     - Subsequent Turns: FlowKV "isolates" (freezes) $C_1$. Because the sink tokens are already preserved in $C_1$, they remain available to the model for all future turns.
> >     - New Turns: For the new incoming turn, we apply the StreamingLLM algorithm again locally to compress the new segment, ensuring any new local sinks are handled if the algorithm requires it. This allows FlowKV to inherit the benefits of the base method (attention sinks) while adding the benefit of isolation (no re-compression of the middle history).
> >
> > ## Question 3: Applicability to long context or long generation tasks.
> > Regarding Long Context (Prefilling): We wish to clarify that FlowKV is primarily optimized for iterative interactions where historical context is repeatedly re-processed (and thus potentially re-compressed) across turns.
> > - In a standard "single-pass" long-context scenario (e.g., summarizing a book in one go), the context is typically processed linearly and compressed once, meaning the "cumulative re-compression" issue FlowKV addresses is less prevalent.
> > - However, in real-world applications, long-context tasks often evolve into "Interactive Long-Context" (e.g., Document QA, "Chat with PDF," or Repository-level Coding). In these scenarios, the long document acts as a static history that serves multiple subsequent queries. Here, FlowKV is highly applicable: it effectively "isolates" the compressed representation of the long document after the first turn, preventing it from degrading as the user asks continuous questions about it.
> >
> > Regarding Long Generation: For long generation tasks, FlowKV contributes to stability by protecting the instructional context.
> > - As the model generates a long response, the KV cache grows, and standard eviction policies might accidentally compress or evict parts of the initial system prompt or user instruction to make room for new tokens.
> > - By "isolating" and locking the prompt's compressed state at the start, FlowKV ensures that the original instruction remains intact throughout the entire generation process. This helps prevents the model from "forgetting" its initial constraints (e.g., formatting rules) during extended generation, as observed in our qualitative case studies (Figure 6).
> > ## Question 4: Performance on larger models (e.g., 32B).
> > Due to computational resource constraints, we focused our extensive ablation studies on 7B and 8B models. However, we firmly believe the conclusions hold for larger models:
> > - Consistent Mechanism: The problem FlowKV solves (cumulative compression error) is mathematical, not scale-dependent. Larger models use the same attention mechanism and suffer from the same "re-compression" loss.
> > - Likely Higher Gain: Larger models typically have stronger capabilities in capturing long-term dependencies. Therefore, they are arguably more sensitive to the loss of historical details caused by naive eviction. We expect FlowKV to show even greater relative gains on 32B+ models by preserving the fine-grained details that these larger models are capable of utilizing.

---

### Author Response · Authors · 2025-11-25
**General Response**

# General Response: Clarifying Memory Budget, Theoretical Updates, and Core Contributions

We sincerely thank the reviewers (eK29, SmkS, CpSM, tsH7) for their constructive feedback and for recognizing the significance of the multi-turn coherence problem.

While we are encouraged by the recognition of our empirical results (boosting **PrefEval retention from ~10% to ~75%**), we noticed shared concerns regarding **memory overhead** and **theoretical depth**. We provide a unified clarification below.

**1. CRITICAL CLARIFICATION: Strict Budget Adherence (No Memory Increase)**
A primary concern (Reviewer CpSM) was that FlowKV "increases the required cache size."
* **Correction:** This is a factual misunderstanding. FlowKV operates under a **strictly fixed global token budget** (e.g., compression ratio $= 0.5$).
* **Mechanism:** We do not simply append new data. Instead, we dynamically allocate the budget: if we preserve historical segments, we compress the *new* turn more aggressively to fit the remaining space.
* **Result:** FlowKV consumes exactly the same memory as baselines (e.g., SnapKV, H2O). The performance gains come from **better bit allocation** (preserving frozen history vs. compressing new noise), not from using more bits.

**2. Novelty: From "Stateless" to "State-Aware" Compression**
Some reviews (SmkS, tsH7) characterized FlowKV as an "engineering adaptation." We respectfully argue that our contribution lies in identifying a fundamental failure mode in SOTA methods.
* **The Flaw:** Prior methods implicitly model KV compression as a **Stateless Operation**, reapplying the same filter function recursively. We prove this causes **Exponential Signal Decay**.
* **The Shift:** FlowKV introduces **State-Aware Isolation**. By acknowledging that "past history is immutable," we shift the paradigm from *Recursive Intersection* to *Piecewise Union*. The simplicity of the solution underscores its robustness, recovering ~65% absolute performance where complex methods fail.

**3. New Theoretical Analysis (Appendix D)**
Addressing requests for rigor (tsH7, CpSM), we have added a **Set-Theoretic Analysis** in the revision.
* We formally show that baseline methods act as a **recursive intersection** of relevant token sets ($S_1 \cap S_2 \dots$), which mathematically approaches an empty set (catastrophic forgetting).
* FlowKV maintains the **union** of relevant sets ($\cup S_i$), providing a formal guarantee that the support set for the prompt remains invariant against query drift.

**4. Additional Experiments**
* **Long Context:** We added **SCBench (Code RepoQA)** results (Response to eK29/CpSM), showing FlowKV effectively handles 64k+ context lengths.

We hope these clarifications demonstrate that FlowKV offers both significant empirical value and solid theoretical grounding.

---

### Note · Authors · 2025-12-18

I have read and agree with the venue's withdrawal policy on behalf of myself and my co-authors.